# Variational autoencoder with weighted samples for high-dimensional non-parametric adaptive importance sampling

**Julien Demange-Chryst**                                    *julien.demange-chryst@onera.fr*
*ONERA/DTIS, Université de Toulouse, F-31055 Toulouse, France*
*Institut de Mathématiques de Toulouse, UMR5219 CNRS, 31062 Toulouse, France*

**François Bachoc**                                    *francois.bachoc@math.univ-toulouse.fr*
*Institut de Mathématiques de Toulouse, UMR5219 CNRS, 31062 Toulouse, France*

**Jérôme Morio**                                    *jerome.morio@onera.fr*
*ONERA/DTIS, Université de Toulouse, F-31055 Toulouse, France*

**Timothé Krauth**                                    *timothe.krauth@zhaw.ch*
*ONERA/DTIS, Université de Toulouse, F-31055 Toulouse, France*
*Zurich University of Applied Sciences, Centre for Aviation, Winterthur, Switzerland*

**Reviewed on OpenReview:** *https://openreview.net/forum?id=nzG9KGssSe*

## Abstract

Adaptive importance sampling is a well-known family of algorithms for density approximation, generation and Monte Carlo integration including rare event estimation. The main common denominator of this family of algorithms is to perform density estimation with weighted samples at each iteration. However, the classical existing methods to do so, such as kernel smoothing or approximation by a Gaussian distribution, suffer from the curse of dimensionality and/or a lack of flexibility. Both are limitations in high dimension and when we do not have any prior knowledge on the form of the target distribution, such as its number of modes. Variational autoencoders are probabilistic tools able to represent with fidelity high-dimensional data in a lower dimensional space. They constitute a parametric family of distributions robust faced to the dimension and since they are based on deep neural networks, they are flexible enough to be considered as non-parametric models. In this paper, we propose to use a variational autoencoder as the auxiliary importance sampling distribution by extending the existing framework to weighted samples. We integrate the proposed procedure in existing adaptive importance sampling algorithms and we illustrate its practical interest on diverse examples.

## 1 Introduction

Importance sampling is a well-known uncertainty quantification method which requires to deal with weighted samples, i.e. a set of observations paired with a set of corresponding weights. It is classically used for estimating an expectation, such as a failure probability, in the aim of reducing the variance of the Monte Carlo estimator (Kahn & Harris, 1951; Rubinstein & Kroese, 2004; Kurtz & Song, 2013), or for generating points from a target probability distribution (Cappé et al., 2004; Cornuet et al., 2012; Marin et al., 2019). What these algorithms have in common is that they all require to estimate a target probability distribution with weighted samples, and obviously, the accuracy of the algorithm depends on the quality of the estimation of the distribution (Chatterjee & Diaconis, 2018). A first way for this estimation is to use non-parametric models, such as kernel smoothing (Wand & Jones, 1994; Scott & Sain, 2005). These models are flexible but they strongly suffer from the curse of dimensionality. Another solution is to use parametric families of distributions, such as the Gaussian (Rubinstein & Kroese, 2004) or Gaussian mixture (Kurtz & Song,

2013; Geyer et al., 2019) ones, which are more robust in medium-high dimension. However, they sometimes require some prior knowledge on the target distribution, and their lack of flexibility and the huge number of parameters to estimate (Au & Beck, 2003) can negatively impact the quality of the estimation when the dimension is high.

In order to combine both flexibility and robustness to the dimension, we suggest here to use as the auxiliary importance sampling distribution a distribution parameterised by a variational autoencoder, whose main principle has been introduced in the last decade (Kingma & Welling, 2014; Kingma et al., 2019). Variational autoencoders are deep generative models for approximating high-dimensional complex distributions of observed data and generating new samples. The specific feature of a variational autoencoder compared to other density estimation methods is that it performs a dimensionality reduction into a lower dimensional latent space in order to facilitate the estimation. Moreover, in opposition to other dimensionality reduction techniques such as principal component analysis (Wold et al., 1987) or autoencoders (McClelland et al., 1987; Bank et al., 2023), variational autoencoders have good generation properties and give explicitly the approximating distribution, allowing to perform Monte Carlo simulations. This tool is now popular in the machine leaning community but not so much in uncertainty quantification.

In the present article, we extend the existing framework to the case of weighted samples by introducing a new objective function. The flexibility of the obtained family of distributions makes it as expressive as a non-parametric model, and despite the very high number of parameters to estimate, this family is much more efficient in high dimension than the classical Gaussian or Gaussian mixture families. Moreover, in order to add flexibility to the model and to be able to learn multimodal distributions, we consider a learnable prior distribution for the variational autoencoder latent variables. We also introduce a new pre-training procedure for the variational autoencoder to find good starting weights of the neural networks to prevent as much as possible the posterior collapse phenomenon to happen. At last, we explicit how the resulting distribution can be combined with importance sampling, and we exploit the proposed procedure in existing adaptive importance sampling algorithms to draw points from a target distribution and to estimate a rare event probability in high dimension on two multimodal problems.

The remainder of this paper is organized as follows. First, Section 2 provides background on importance sampling. Section 3 formally presents the problem of density estimation and provides a review on the principle of variational autoencoders and some improvements. Then, Section 4 introduces our suggested extension of variational autoencoders to weighted samples. In addition, Section 5 presents the posterior collapse phenomenon and a procedure to handle it. Section 6 illustrates the practical interest of the proposed procedure on the generation from some target distributions and on the estimation of failure probabilities on two multimodal problems. Finally, Section 7 concludes the present article and gives future research perspectives stemming from it.

## 2 Background on importance sampling

In this section, we recall the main principle of importance sampling as well as some important requirements. Then, we explicit the main challenges of importance sampling in high dimension and some existing remedies to make the method more accurate. At last, we make a brief review of existing methods using neural networks for Monte Carlo simulations and adaptive importance sampling.

### 2.1 General presentation of importance sampling

Importance sampling (IS) is a classical variance-reduction technique which was introduced in (Kahn & Harris, 1951; Hammersley & Handscomb, 1961) and massively used for rare event estimation (Shinozuka, 1983; Harbitz, 1986; Papaioannou et al., 2019) in particular. In the case of the estimation of an expectation $I = \mathbb{E}_f(\psi(\mathbf{X}))$ where $\psi : \mathcal{X} \subseteq \mathbb{R}^d \longrightarrow \mathbb{R}$ is a black-box function, it consists in rewriting the expectation according to an auxiliary density $g : \mathcal{X} \longrightarrow \mathbb{R}_+$ as $\mathbb{E}_g(\psi(\mathbf{X}) w^g(\mathbf{X}))$, where $w^g(\mathbf{x}) = f(\mathbf{x})/g(\mathbf{x})$ is the likelihood ratio. To get an unbiased estimate, the support of $g$ must contain the support of $\mathbf{x} \in \mathcal{X} \mapsto$

$\psi(\mathbf{x}) f(\mathbf{x})$. The corresponding IS estimator is then given by:

$$\widehat{I}_{g,N}^{\mathrm{IS}} = \frac{1}{N} \sum_{n=1}^{N} \psi\left(\mathbf{X}^{(n)}\right) w^g\left(\mathbf{X}^{(n)}\right), \tag{1}$$

where $\left(\mathbf{X}^{(n)}\right)_{n \in [\![1,N]\!]}$ is an i.i.d. sample distributed according to the IS auxiliary distribution $g$. It is consistent and unbiased, and it has zero-variance if and only if $g = g_{\mathrm{opt}}$ with $\forall \mathbf{x} \in \mathcal{X}$, $g_{\mathrm{opt}}(\mathbf{x}) \propto \psi(\mathbf{x}) f(\mathbf{x})$ (Bucklew, 2004), on the condition that $\psi$ is non-negative. However, this optimal density cannot be used in practice because the normalizing constant is $I$, which is the quantity to estimate. Nevertheless, one can still expect a significant variance reduction by approximating the optimal density by a sub-optimal auxiliary distribution $g$ close to $g_{\mathrm{opt}}$ in a certain sense (more details will be given in Section 2.2). Moreover, importance sampling can also be used in a Bayesian context for generating samples according to a target distribution known up to a constant (Cappé et al., 2004; Cornuet et al., 2012; Marin et al., 2019; Martino et al., 2015; El-Laham & Bugallo, 2019; 2021). We refer to (Bugallo et al., 2017; Elvira & Martino, 2021) for a more exhaustive presentation of classical importance sampling algorithms.

The common denominator of every importance sampling procedure is that we need to be able to not only sample from the built auxiliary distribution, but also to have access to its PDF values. In the specific case of the estimation of an expectation, it is crucial to get them to be able to compute the importance sampling estimator in equation 1. Moreover, in adaptive importance sampling algorithms, the likelihood ratios are required at the end of an iteration to be able to perform the next one.

## 2.2 High-dimensional importance sampling

As briefly said in the previous sub-section, the main challenge of every importance sampling method is to approximate a target distribution by an auxiliary one. As first way to do so is to use non-parametric approximating distributions. Various methods based on kernel smoothing with (Zhang, 1996; Morio, 2011; 2012) and without (Ang et al., 1992; West, 1993; Wand & Jones, 1994) weighted samples have been investigated. They are flexible enough to be adapted to a large range of problems, especially multimodal problems. However, they strongly suffer from the curse of dimensionality since the size of the required sample to have a good approximation of the target distribution exponentially grows with the dimension and despite some improvements (Silverman, 1986; Zhang et al., 2006; Perrin et al., 2018), these models are very inaccurate in high dimension.

Second, a target distribution can also be approximated within a parametric family of distributions. The most classical way to choose the best candidate is to minimise the Kullback-Leibler divergence (Kullback & Leibler, 1951) between the target distribution and the candidate one w.r.t. the parameters of the family. This choice is relevant since a relation between the value of the Kullback-Leibler divergence and the variance of the importance sampling estimator has been established in (Chatterjee & Diaconis, 2018).

The most common and convenient parametric auxiliary families are the *Gaussian* family (Rubinstein & Kroese, 2004; De Boer et al., 2005) and the *Gaussian mixture* family (Kurtz & Song, 2013; Geyer et al., 2019) for multimodal target distribution. However, the amount of parameters to estimate for these distributions makes them inaccurate when the dimension increases, as well as when there is a lack of a prior knowledge on the form of the target distribution such as its number of modes for example. An inaccurate approximation of the target distribution can lead to the weight degeneracy phenomenon (Rubinstein & Glynn, 2009), when a few number of likelihood ratios dominates the other ones, and which strongly badly affects the performances of any importance sampling algorithm. Numerical techniques have been investigated in order to make the density approximation more robust faced to the dimension, such as methods to control the likelihood ratios (Ionides, 2008; Koblents & Míguez, 2015; Martino et al., 2018), shrinkage of the covariance matrix (Ledoit & Wolf, 2004; El-Laham et al., 2019) or methods to reduce the effective dimension of the problem (Rubinstein & Glynn, 2009; Uribe et al., 2021; El Masri et al., 2021). A final method is to consider families of distributions more adapted to high dimension. In the specific case where the input distribution is the standard Gaussian distribution, the *von Mises–Fisher–Nakagami* (vMFNM) family of distributions (Mardia & Jupp, 2000; Wang & Song, 2016; Papaioannou et al., 2019) is well-suited and more robust when the dimension is high. However,

when considering mixtures of vMFNM distributions for multimodal target distributions, the corresponding algorithms require the knowledge of the number of modes.

## 2.3 Monte Carlo simulations and importance sampling with neural networks

The use of neural networks for Monte Carlo analyses has also already been investigated in the literature. First, the authors of (Papadrakakis et al., 1996; Hurtado & Alvarez, 2001; Papadrakakis & Lagaros, 2002) apply neural networks for Monte Carlo simulation and risk analysis. Moreover, the authors of (Naesseth et al., 2018; Le et al., 2018; Lawson et al., 2018) use latent-variable models as approximating models for sequential Monte Carlo by maximising a lower bound of the log-likelihood to be more efficient in high dimension. Then, the use of normalising flows for high-dimensional Monte Carlo integration and sampling (Müller et al., 2019; Gao et al., 2020; Arbel et al., 2021) have been proposed. In addition, the idea of importance sampling has also been considered to propose a more accurate objective function for some latent-variable models (Burda et al., 2016).

More related to our work, the authors of (Dowling et al., 2018) combine adaptive importance sampling and variational inference whereas the authors of (Wang et al., 2019) propose an adaptive importance sampling algorithm supported by a variational autoencoder to sample from a target distribution known up to a constant. At last, the authors of (Dasgupta & Johnson, 2024) introduce a new algorithm coupling normalizing flows and importance sampling for the estimation of rare event probabilities in high dimension and without any prior knowledge on the form of the failure domain.

## 3 Probability density estimation supported by variational autoencoders

Probability density function (PDF) estimation is a major topic of interest in statistics, as the PDF fully characterises the distribution of a continuous random vector. Given an observed dataset $\left(\mathbf{X}^{(n)}\right)_{n \in [\![1,N]\!]}$ of points from the *input space* $\mathcal{X} \subseteq \mathbb{R}^d$ of dimension $d \geq 1$, it consists in the construction of an estimate of the true underlying unknown PDF $g^* : \mathcal{X} \to \mathbb{R}_+$. There exist several strategies to do so. First, parametric methods consist in picking the best representative of the target density $g^*$ within a parametric family of densities. A few classical families (Kotz et al., 2004) are often used and can accurately model a fair number of phenomena, but their lack of flexibility and/or the high amount of parameters to estimate when the dimension $d$ increases is a limitation. Second, an important class of non-parametric methods is provided by kernel smoothing (Wand & Jones, 1994; Scott & Sain, 2005). Despite the larger flexibility of the corresponding approximating models and some improvements (Silverman, 1986; Zhang et al., 2006; Perrin et al., 2018), kernel smoothing strongly suffers from the curse of dimensionality. One can also mention functional decomposition methods (Wasserman, 2006) or non-parametric Bayesian methods (Ghosal & van der Vaart, 2017), but they also suffer from the curse of dimensionality.

A recent strategy to perform probability density estimation in high dimension is offered by variational autoencoders (VAEs). The principle of this probabilistic tool was first introduced in (Kingma & Welling, 2014). We refer to (Kingma et al., 2019) for an exhaustive and up-to-date presentation. Let us then present in this section the general principle of VAEs.

### 3.1 Latent variable and Bayesian variational inference

Bayesian variational inference (Fox & Roberts, 2012) consists in approximating the target distribution $g^*$ by a parametric model $g_{\boldsymbol{\theta}}$, where $\boldsymbol{\theta}$ denotes its parameters. In order to facilitate the density estimation, Bayesian variational inference introduces an unobserved *latent variable* $\mathbf{z}$ which lies in a lower-dimensional space $\mathcal{Z} \subseteq \mathbb{R}^{d_z}$, with $d_z \ll d$. Together, both variables define a joint distribution on $\mathcal{X} \times \mathcal{Z}$. By marginalising over the latent variable $\mathbf{z}$, the distribution on $\mathcal{X}$ is given by:

$$g_{\boldsymbol{\theta}}\left(\mathbf{x}\right) = \int_{\mathcal{Z}} g_{\boldsymbol{\theta}}\left(\mathbf{x}, \mathbf{z}\right) d\mathbf{z}. \tag{2}$$

Nevertheless, the marginalisation over an unknown and unobserved latent variable makes the computation of the integral intractable, and as a consequence also the direct computation of the estimated distribution

$g_{\boldsymbol{\theta}}$. However, thanks to Bayes theorem, it is possible to write:

$$g_{\boldsymbol{\theta}}\left(\mathbf{z}\,|\mathbf{x}\right)=\frac{p\left(\mathbf{z}\right)g_{\boldsymbol{\theta}}\left(\mathbf{x}\,|\mathbf{z}\right)}{g_{\boldsymbol{\theta}}\left(\mathbf{x}\right)}, \tag{3}$$

where $p$ is the prior distribution on $\mathbf{z}$ and where $g_{\boldsymbol{\theta}}\left(\mathbf{x}\,|\mathbf{z}\right)$ is the likelihood. Being able to infer the *true posterior* $g_{\boldsymbol{\theta}}\left(\mathbf{z}\,|\mathbf{x}\right)$ allows then to compute $g_{\boldsymbol{\theta}}\left(\mathbf{x}\right)$ from equation 3. To do so, we approximate the true posterior by a *variational posterior distribution* $q_{\boldsymbol{\phi}}\left(\mathbf{z}\,|\mathbf{x}\right)$ chosen within an expressive parametric family of distributions parameterized by $\boldsymbol{\phi}$.

The choice of the dimension of the latent space $d_z$ has a huge influence on the accuracy of the density approximation and should be the best trade-off between dimensionality reduction and loss of information. Indeed, performing the variational inference in a lower dimensional subspace $\mathcal{Z}$ instead of in the high-dimensional input space $\mathcal{X}$ reduces the number of parameters of $q_{\boldsymbol{\phi}}\left(\mathbf{z}\,|\mathbf{x}\right)$ to estimate and makes the process more accurate. However, the dimension of the latent space must be large enough to correctly catch the structure of the data, and thus the structure of the true underlying distribution $g^*$.

## 3.2 General principle of VAEs

In order to perform the density estimation, a VAE consists in using two neural networks, a probabilistic encoder denoted $E_{\boldsymbol{\phi}}$ parameterized by the weights $\boldsymbol{\phi}$, and a probabilistic decoder denoted $D_{\boldsymbol{\theta}}$ parameterized by the weights $\boldsymbol{\theta}$, to model respectively the variational posterior distribution and the likelihood function. Hence, in the case of VAEs, the parameters $\boldsymbol{\phi}$ and $\boldsymbol{\theta}$ discussed in Section 3.1 are the weights of $E_{\boldsymbol{\phi}}$ and $D_{\boldsymbol{\theta}}$.

First, the probabilistic encoder performs the dimensionality reduction described above. Indeed, given an input point from the input space $\mathbf{x} \in \mathcal{X}$, it returns the parameters of the approximating variational parametric distribution $q_{\boldsymbol{\phi}}\left(\mathbf{z}\,|\mathbf{x}\right)$. Classically, a Gaussian distribution with diagonal covariance matrix is considered as the variational posterior distribution, such that $q_{\boldsymbol{\phi}}\left(.\,|\mathbf{x}\right) \sim \mathcal{N}\left(\boldsymbol{\mu}_{\mathbf{x}}, \boldsymbol{\Sigma}_{\mathbf{x}}\right)$, where its parameters $\left(\boldsymbol{\mu}_{\mathbf{x}}, \boldsymbol{\Sigma}_{\mathbf{x}}\right) = E_{\boldsymbol{\phi}}\left(\mathbf{x}\right)$ are the output of the encoder.

Second, the probabilistic decoder sets the parameters of the likelihood function $g_{\boldsymbol{\theta}}\left(\mathbf{x}\,|\mathbf{z}\right)$. Indeed, it maps a point from the latent space $\mathbf{z} \in \mathcal{Z}$ into the corresponding parameters of the likelihood. Once more, for continuous data, the most classical choice for the likelihood is the Gaussian distribution with diagonal covariance matrix. Thus, the likelihood distribution is given by $g_{\boldsymbol{\theta}}\left(.\,|\mathbf{z}\right) \sim \mathcal{N}\left(\boldsymbol{\mu}_{\mathbf{z}}, \boldsymbol{\Sigma}_{\mathbf{z}}\right)$, where its parameters $\left(\boldsymbol{\mu}_{\mathbf{z}}, \boldsymbol{\Sigma}_{\mathbf{z}}\right) = D_{\boldsymbol{\theta}}\left(\mathbf{z}\right)$ are the output of the decoder. As a result, according to equation 2 and with $g_{\boldsymbol{\theta}}\left(\mathbf{x}, \mathbf{z}\right) = p\left(\mathbf{z}\right)g_{\boldsymbol{\theta}}\left(\mathbf{x}\,|\mathbf{z}\right)$, the estimated distribution $g_{\boldsymbol{\theta}}$ can be seen as an infinite mixture of Gaussian distributions.

Even if the distribution estimated by a VAE belongs theoretically to a parametric family, its flexibility, its ability to approximate complex distributions and its form make it closer to a non-parametric model. As discussed at the beginning of Section 3, other existing non-parametric models strongly face the curse of dimensionality whereas, as shown in the numerical results in Section 6, VAEs are much more efficient in high dimension.

The main challenge now is to train both neural networks. The most common method for this is to minimize the Kullback-Leibler divergence (Kullback & Leibler, 1951) $D_{\mathrm{KL}}\left(g^*\|g_{\boldsymbol{\theta}}\right)$ between the target distribution $g^*$ and the candidate one $g_{\boldsymbol{\theta}}$ w.r.t. the parameters $\boldsymbol{\theta}$. One can show that it is equivalent to maximise the expectation $\mathbb{E}_{g^*}\left[\log\left(g_{\boldsymbol{\theta}}\left(\mathbf{X}\right)\right)\right]$. This is also called cross-entropy minimisation (Rubinstein & Kroese, 2004). Nevertheless, the computation of this expectation requires the computation of the integral in equation 2 which is intractable. However, for fixed $\mathbf{x} \in \mathcal{X}$, one can find a more convenient and easier to compute lower bound of the log-likelihood using the latent variable $\mathbf{z}$ (Kingma & Welling, 2014; Kingma et al., 2019):

$$\log\left(g_{\boldsymbol{\theta}}\left(\mathbf{x}\right)\right) \geq \mathbb{E}_{q_{\boldsymbol{\phi}}(.|\mathbf{x})}\left[\log\left(g_{\boldsymbol{\theta}}\left(\mathbf{x}|\mathbf{Z}\right)\right)\right] - D_{\mathrm{KL}}\left(q_{\boldsymbol{\phi}}\left(\mathbf{Z}|\mathbf{x}\right)\|p\left(\mathbf{Z}\right)\right). \tag{4}$$

Taking the expectation of $\mathbf{x}$ w.r.t. $g^*$ leads to the objective function of the VAE given by:

$$\mathbb{E}_{g^*}\left[\log\left(g_{\boldsymbol{\theta}}\left(\mathbf{X}\right)\right)\right] \geq \mathbb{E}_{g^*}\left[\mathbb{E}_{q_{\boldsymbol{\phi}}(.|\mathbf{X})}\left[\log\left(g_{\boldsymbol{\theta}}\left(\mathbf{X}|\mathbf{Z}\right)\right)\right]\right] - \mathbb{E}_{g^*}\left[D_{\mathrm{KL}}\left(q_{\boldsymbol{\phi}}\left(\mathbf{Z}|\mathbf{X}\right)\|p\left(\mathbf{Z}\right)\right)\right] := \mathrm{ELBO}\left(\boldsymbol{\phi}, \boldsymbol{\theta}\right). \tag{5}$$

The training procedure of the VAE aims then to maximize the ELBO (*Evidence lower bound*) function according to the weights $\left(\boldsymbol{\phi}, \boldsymbol{\theta}\right)$. In practice, all the terms of this objective function are estimated using the

observed sample $\left(\mathbf{X}^{(n)}\right)_{n \in [\![1,N]\!]}$ distributed according to $g^*$. More precisely, the ELBO function combines two opposite phenomena, and optimising it consists in finding the best trade-off between them.

- First, maximising the reconstruction term $\mathbb{E}_{q_\phi(.|\mathbf{x})}\left[\log\left(g_{\boldsymbol{\theta}}\left(\mathbf{x}|\mathbf{Z}\right)\right)\right]$ in order to have an accurate reconstruction. More precisely, the encoded distributions $q_\phi\left(.|\mathbf{X}^{(n)}\right)$ associated to each point from the dataset must be separated enough from each others in the latent space such that the decoder is able to reconstruct the correct point $\mathbf{X}^{(n)}$ from it.

- Second, minimising the regularisation term, which is the Kullback-Leibler divergence $D_{\mathrm{KL}}\left(q_\phi\left(\mathbf{Z}|\mathbf{x}\right)\|p\left(\mathbf{Z}\right)\right)$ between the prior and the variational posterior distribution over the dataset. Indeed, the decoder is trained from samples distributed as $q_\phi\left(.|\mathbf{X}^{(n)}\right)$ for all $n \in [\![1,N]\!]$. Then, when new $\mathbf{x}$ samples are generated, the decoder is applied to new $\mathbf{z}$ samples distributed according to the prior $p$. If this prior is too far away from all the $q_\phi\left(.|\mathbf{X}^{(n)}\right)$, the new generated points may not be representative of the true underlying distribution $g^*$, because the decoder is applied outside of its training regime.

### 3.3 Choice of the prior and VampPrior

Let us discuss the choice of the prior distribution $p$ on $\mathbf{z}$. Classically, the most common choice for the prior is the standard Normal distribution in dimension $d_z$. This simple choice is practically convenient, especially because it gives an analytical expression of the Kullback-Leibler divergence $D_{\mathrm{KL}}\left(q_\phi\left(\mathbf{Z}|\mathbf{x}\right)\|p\left(\mathbf{Z}\right)\right)$ since $q_\phi\left(\mathbf{z}|\mathbf{x}\right)$ is also Gaussian. However, it is not optimal in general, for multimodal problems for example, and can lead to over-regularisation and consequently to poor density estimation performances. A more flexible prior is most of the time required.

Theoretically, by rewriting the ELBO function with two regularisation terms and by maximising it w.r.t. the prior $p$ (Makhzani et al., 2015; Hoffman & Johnson, 2016), the optimal prior is analytically given by:

$$p^*\left(\mathbf{z}\right) = \int_{\mathcal{X}} q_\phi\left(\mathbf{z}|\mathbf{x}\right) g^*\left(\mathbf{x}\right) d\mathbf{x} = \mathbb{E}_{g^*}\left[q_\phi\left(\mathbf{z}|\mathbf{X}\right)\right]. \tag{6}$$

This optimal prior distribution is called the *aggregated posterior*. One can note that the aggregated posterior is the continuous mixture of all the variational posterior distributions over the whole input space. In particular, it depends on the parameters of the encoder $\phi$. A natural way to approximate $p^*$ is $p^{\mathrm{emp}}\left(\mathbf{z}\right) = \frac{1}{N}\sum_{n=1}^{N} q_\phi\left(\mathbf{z}|\mathbf{X}^{(n)}\right)$. Nevertheless, since $N$ is usually large, this empirical distribution is intractable in practice and can also lead to over-fitting. To overcome these issues, there are several types of priors to approximate the optimal one $p^*$ (Dilokthanakul et al., 2016; Nalisnick et al., 2016; Kim & Mnih, 2018; Chen et al., 2018; Lavda et al., 2019; Takahashi et al., 2019; Kalatzis et al., 2020; Negri et al., 2022), each of them with advantages and drawbacks. Here, we use the *Variational Mixture of Posteriors* prior, or *VampPrior*, introduced in (Tomczak & Welling, 2018). It consists in approximating the optimal prior by a mixture distribution of the form:

$$p_{\mathbf{u}_1,\ldots,\mathbf{u}_K,\phi}^{\mathrm{VP}}\left(\mathbf{z}\right) = \frac{1}{K}\sum_{k=1}^{K} q_\phi\left(\mathbf{z}|\mathbf{u}_k\right), \tag{7}$$

where $K \geq 1$ is the number of components of the mixture chosen by the user and where the points $\left(\mathbf{u}_k\right)_{k\in[\![1,K]\!]} \in \mathcal{X}$ are learnable pseudo-inputs from the input space. More precisely, the VampPrior distributions with $K$ components constitute a parametric family of distributions parameterised by $\left(\mathbf{u}_1,\ldots,\mathbf{u}_K,\phi\right)$.

Practically, a single neural network denoted $\mathrm{VP}_{\boldsymbol{\lambda}}$ and parameterised by the weights $\boldsymbol{\lambda}$ maps the $K$ vectors of the canonical basis $\left(e_k^K\right)_{k\in[\![1,K]\!]}$ of $\mathbb{R}^K$ into the pseudo-inputs $\left(\mathbf{u}_k\right)_{k\in[\![1,K]\!]}$. Therefore, the VampPrior distribution is only parameterised by the weights $\boldsymbol{\lambda}$ and $\phi$. This new neural network has to be trained in order to find the best pseudo-inputs that maximise the performances of the VAE, and these new parameters have then to appear in the ELBO function which thus becomes:

$$\mathrm{ELBO}\left(\phi,\boldsymbol{\theta},\boldsymbol{\lambda}\right) = \mathbb{E}_{g^*}\left[\mathbb{E}_{q_\phi(.|\mathbf{X})}\left[\log\left(g_{\boldsymbol{\theta}}\left(\mathbf{X}|\mathbf{Z}\right)\right)\right]\right] - \mathbb{E}_{g^*}\left[D_{\mathrm{KL}}\left(q_\phi\left(\mathbf{Z}|\mathbf{X}\right)\|p_{\boldsymbol{\lambda},\phi}\left(\mathbf{Z}\right)\right)\right], \tag{8}$$

where $p_{\boldsymbol{\lambda},\boldsymbol{\phi}}^{\mathrm{VP}}(\mathbf{z}) = \frac{1}{K}\sum_{k=1}^{K} q_{\boldsymbol{\phi}}\left(\mathbf{z}|\mathrm{VP}_{\boldsymbol{\lambda}}\left(\mathbf{e}_k^K\right)\right)$. The VampPrior is flexible enough to be adapted to many kinds of problems and has the advantage to depend on the weights of the encoder $\boldsymbol{\phi}$, as in the form of the optimal prior in equation 6.

## 4 Variational autoencoder with weighted samples

In this section, we present a new procedure to estimate, with a VAE, a target probability density $g^*$, no longer by using observations drawn from $g^*$ itself, but rather by using observations drawn from another probability density $f : \mathcal{X} \to \mathbb{R}_+$. To the best of our knowledge, this problem has only been investigated in (Wang et al., 2019) (see Remark 5.1 and Section 6 for comparisons). To that end, we show how to adapt the VAE framework presented in Section 3 to address this issue.

In the context of the rest of the paper, we assume that we have at our disposal a dataset $\left(\mathbf{X}^{(n)}\right)_{n\in[\![1,N]\!]} \in \mathcal{X}^N$ distributed according to a distribution $f : \mathcal{X} \to \mathbb{R}_+$ known up to a constant, and that we aim to estimate the target probability density $g^*$ also known up to a constant. More precisely, the distributions $f$ and $g^*$ are given by:

$$\begin{cases} f(\mathbf{x}) &= \widetilde{f}(\mathbf{x})/c_f \\ g^*(\mathbf{x}) &= \widetilde{g}^*(\mathbf{x})/c_g, \end{cases} \tag{9}$$

where $\widetilde{f}$ and $\widetilde{g}^*$ are fully known non-negative functions and where $c_f$ and $c_g$ are positive constants. At last, we also assume that the support of $f$ contains the support of $g^*$.

In the same way as in Section 3, in order to approximate as accurately as possible the target distribution $g^*$, we aim to minimise the Kullback-Leibler divergence $D_{\mathrm{KL}}\left(g^*\|g_{\boldsymbol{\theta}}\right)$ w.r.t. the parameters $\boldsymbol{\theta}$, which is equivalent to maximise $\mathbb{E}_{g^*}\left[\log\left(g_{\boldsymbol{\theta}}(\mathbf{X})\right)\right]$. However, we have here a sample distributed according to $f$ and it is not possible to compute this expectation under $g^*$. Fortunately, thanks to the importance sampling trick (Kahn & Harris, 1951), it is possible to rewrite it as an expectation under $f$:

$$\mathbb{E}_{g^*}\left[\log\left(g_{\boldsymbol{\theta}}(\mathbf{X})\right)\right] = \mathbb{E}_f\left[\frac{g^*(\mathbf{X})}{f(\mathbf{X})}\log\left(g_{\boldsymbol{\theta}}(\mathbf{X})\right)\right]. \tag{10}$$

The latter expectation still requires the computation of the intractable integral of $g_{\boldsymbol{\theta}}$ in equation 2. Nevertheless, by multiplying both sides of equation 4 by the positive weight $g^*(\mathbf{x})/f(\mathbf{x})$, we can obtain a computable lower bound of the weighted log-likelihood depending on the latent variable $\mathbf{z}$:

$$\frac{g^*(\mathbf{x})}{f(\mathbf{x})}\log\left(g_{\boldsymbol{\theta}}(\mathbf{x})\right) \geq \frac{g^*(\mathbf{x})}{f(\mathbf{x})}\left(\mathbb{E}_{q_{\boldsymbol{\phi}}(.|\mathbf{x})}\left[\log\left(g_{\boldsymbol{\theta}}(\mathbf{x}|\mathbf{Z})\right)\right] - D_{\mathrm{KL}}\left(q_{\boldsymbol{\phi}}(\mathbf{Z}|\mathbf{x})\|p_{\boldsymbol{\lambda},\boldsymbol{\phi}}(\mathbf{Z})\right)\right). \tag{11}$$

Finally, applying the expectation w.r.t. $\mathbf{X} \sim f$, we get:

$$\begin{aligned} \mathbb{E}_{g^*}\left[\log\left(g_{\boldsymbol{\theta}}(\mathbf{X})\right)\right] &= \mathbb{E}_f\left[\frac{g^*(\mathbf{X})}{f(\mathbf{X})}\log\left(g_{\boldsymbol{\theta}}(\mathbf{X})\right)\right] \\ &\geq \mathbb{E}_f\left[\frac{g^*(\mathbf{X})}{f(\mathbf{X})}\left(\mathbb{E}_{q_{\boldsymbol{\phi}}(.|\mathbf{X})}\left[\log\left(g_{\boldsymbol{\theta}}(\mathbf{X}|\mathbf{Z})\right)\right] - D_{\mathrm{KL}}\left(q_{\boldsymbol{\phi}}(\mathbf{Z}|\mathbf{X})\|p_{\boldsymbol{\lambda},\boldsymbol{\phi}}(\mathbf{Z})\right)\right)\right]. \end{aligned} \tag{12}$$

As in Section 3, the training procedure of the VAE will here aim to maximise the lower bound of equation 12 over the data. At last, even though $f$ and $g^*$ are known only up to positive constants $c_f$ and $c_g$, one can ignore these constants in the expression of the function to maximise. Therefore, the objective function in the current framework is given by the new weighted ELBO function:

$$\mathrm{wELBO}\left(\boldsymbol{\phi},\boldsymbol{\theta},\boldsymbol{\lambda}\right) = \mathbb{E}_f\left[\frac{\widetilde{g}^*(\mathbf{X})}{\widetilde{f}(\mathbf{X})}\mathbb{E}_{q_{\boldsymbol{\phi}}(.|\mathbf{X})}\left[\log\left(g_{\boldsymbol{\theta}}(\mathbf{X}|\mathbf{Z})\right)\right]\right] - \mathbb{E}_f\left[\frac{\widetilde{g}^*(\mathbf{X})}{\widetilde{f}(\mathbf{X})}D_{\mathrm{KL}}\left(q_{\boldsymbol{\phi}}(\mathbf{Z}|\mathbf{X})\|p_{\boldsymbol{\lambda},\boldsymbol{\phi}}(\mathbf{Z})\right)\right]. \tag{13}$$

Maximising $\mathrm{wELBO}\left(\boldsymbol{\phi},\boldsymbol{\theta},\boldsymbol{\lambda}\right)$ w.r.t. the parameters $\boldsymbol{\phi}$, $\boldsymbol{\theta}$ and $\boldsymbol{\lambda}$ is equivalent to maximising the right-hand side of equation 12 w.r.t. $\boldsymbol{\phi}$, $\boldsymbol{\theta}$ and $\boldsymbol{\lambda}$, which is a lower bound of $\mathbb{E}_{g^*}\left[\log\left(g_{\boldsymbol{\theta}}(\mathbf{X})\right)\right]$. To sum up, only

by modifying the objective function of the classical VAE from Section 3, it is possible to learn the high-dimensional target distribution $g^*$ from samples distributed according to an initial distribution $f$ with a VAE. In practice, all the terms of this objective function are estimated using the observed sample $\left(\mathbf{X}^{(n)}\right)_{n \in [\![1,N]\!]}$ distributed according to $f$.

# 5 New initialisation procedure of the weights of the neural networks to prevent posterior collapse

## 5.1 Posterior collapse

A classical problem that badly affects the accuracy of the density estimation and the generating properties of the VAE is *posterior collapse*. As described in many articles (Bowman et al., 2015; Higgins et al., 2016; Sønderby et al., 2016; He et al., 2019), posterior collapse generally refers to an over-regularisation of the VAE, a loss of information, which mathematically traduces that the Kullback-Leibler term of the objective function vanishes, that is $D_{\mathrm{KL}}\left(q_{\boldsymbol{\phi}}\left(\mathbf{Z}|\mathbf{x}\right)\|p_{\boldsymbol{\lambda},\boldsymbol{\phi}}\left(\mathbf{Z}\right)\right) \approx 0$ for every $\mathbf{x} \in \mathcal{X}$. In order words, every variational posterior distribution collapses into the prior. Getting stuck in a local maxima during the optimisation can explain this phenomenon (Sønderby et al., 2016; Lucas et al., 2019). Notably, a trained VAE affected by posterior collapse is not able to catch the different modes of a multimodal distribution. Note that (Takida et al., 2020) investigates an alternative definition of posterior collapse based on the mutual information.

The most classical solution to deal with posterior collapse is to introduce a coefficient $\beta \in [0,1]$ before the Kullback-Leibler term of the objective function in order to reduce the regularisation effect (Bowman et al., 2015; Higgins et al., 2016; Sønderby et al., 2016), but the corresponding objective function is no longer a lower bound of the log-likelihood of the model. Others papers propose an alternative formulation of the objective function (Rezende & Viola, 2018; Alemi et al., 2018) whereas the works of (Razavi et al., 2019; Xu & Durrett, 2018) investigate new choices for the family of distributions for the prior or the variational posterior.

## 5.2 New pre-initialisation procedure

In the present article, we introduce a new solution to handle posterior collapse. As explained above, posterior collapse is caused by getting stuck in a local maximum of the objective function. In order to prevent it, we propose a pre-initialisation procedure of the weights of the neural networks $\boldsymbol{\phi}$, $\boldsymbol{\theta}$ and $\boldsymbol{\lambda}$ in order to start the training of the VAE from more adapted starting points $\boldsymbol{\phi}^{(0)}$, $\boldsymbol{\theta}^{(0)}$ and $\boldsymbol{\lambda}^{(0)}$.

First, we initialise the weights $\boldsymbol{\lambda}$ by supervised learning. To do so, we randomly pick without replacement a sequence of indices $(s(k))_{k \in [\![1,K]\!]}$ of integers of $[\![1, N]\!]$ with probabilities proportional to the family $\left(\widetilde{g}^*\left(\mathbf{X}^{(n)}\right) / \widetilde{f}\left(\mathbf{X}^{(n)}\right)\right)_{n \in [\![1,N]\!]}$ in order to create the sub-sample $\left(\mathbf{X}^{(s(k))}\right)_{k \in [\![1,K]\!]}$. Then, we train the neural network $\mathrm{VP}_{\boldsymbol{\lambda}}$ such that it maps each vector $\mathbf{e}_k$ into the corresponding picked point from the dataset $\mathbf{X}^{(s(k))}$ by minimising the mean square error:

$$\boldsymbol{\lambda}^{(0)} = \arg\min_{\boldsymbol{\lambda}} \sum_{k=1}^{K} \left(\mathrm{VP}_{\boldsymbol{\lambda}}\left(\mathbf{e}_k\right) - \mathbf{X}^{(s(k))}\right)^{\top} \left(\mathrm{VP}_{\boldsymbol{\lambda}}\left(\mathbf{e}_k\right) - \mathbf{X}^{(s(k))}\right). \tag{14}$$

The proposed sampling strategy of the sub-sample of size $K$ ensures that it is distributed according to the target distribution $g^*$ rather than the initial distribution $f$. Therefore, once pre-trained, the initial pseudo-inputs $\boldsymbol{u}_k^{(0)} = \mathrm{VP}_{\boldsymbol{\lambda}^{(0)}}\left(\mathbf{e}_k\right)$ will already be representative of the target distribution $g^*$. As a result, the characteristics of the target distribution $g^*$, such as its modes, will be already caught by $\mathrm{VP}_{\boldsymbol{\lambda}^{(0)}}$.

Second, we initialise the weights $\boldsymbol{\phi}$ and $\boldsymbol{\theta}$ by unsupervised learning. To do so, in order to already ensure good reconstruction properties, we train the pair encoder/decoder $(E_{\boldsymbol{\phi}}, D_{\boldsymbol{\theta}})$ as a classical autoencoder on the whole dataset by minimising the mean square error between the data points and the reconstructed ones (encoded and then decoded). Once more, thanks to the importance sampling trick, we are able to rewrite the loss function as an expectation under $f$. Moreover, since a classical autoencoder only cares about the

mean values $\boldsymbol{\mu_x}$ and $\boldsymbol{\mu_z}$, we add a penalisation term in order to also pre-train the weights of the networks corresponding to the scale parameters of the encoder in order to initialise the diagonal terms of $\boldsymbol{\Sigma_x}$ close to 1 (for normalized data).

We write $\left( E_{\boldsymbol{\phi}}^{\boldsymbol{\mu}}\left(\mathbf{x}\right), E_{\boldsymbol{\phi}}^{\boldsymbol{\Sigma}}\left(\mathbf{x}\right) \right) = E_{\boldsymbol{\phi}}\left(\mathbf{x}\right)$ and $\left( D_{\boldsymbol{\theta}}^{\boldsymbol{\mu}}\left(\mathbf{z}\right), D_{\boldsymbol{\theta}}^{\boldsymbol{\Sigma}}\left(\mathbf{z}\right) \right) = D_{\boldsymbol{\theta}}\left(\mathbf{z}\right)$ and we recall that $E_{\boldsymbol{\phi}}^{\boldsymbol{\Sigma}}\left(\mathbf{x}\right)$ and $D_{\boldsymbol{\theta}}^{\boldsymbol{\Sigma}}\left(\mathbf{z}\right)$ are diagonal matrices of size respectively $d_z \times d_z$ and $d \times d$ representing the output covariance matrices. We also let $\log^2\left(D\right)$ be obtained by taking the square logarithm of a diagonal matrix D, and we get:

$$
\begin{aligned}
\left( \boldsymbol{\phi}^{(0)}, \boldsymbol{\theta}^{(0)} \right) &= \underset{\boldsymbol{\phi}, \boldsymbol{\theta}}{\arg\min}\, \mathbb{E}_{g^*}\left[ \left(\mathbf{X} - D_{\boldsymbol{\theta}}^{\boldsymbol{\mu}}\left(E_{\boldsymbol{\phi}}^{\boldsymbol{\mu}}\left(\mathbf{X}\right)\right)\right)^{\top}\left(\mathbf{X} - D_{\boldsymbol{\theta}}^{\boldsymbol{\mu}}\left(E_{\boldsymbol{\phi}}^{\boldsymbol{\mu}}\left(\mathbf{X}\right)\right)\right) + \frac{1}{d_z}\mathrm{Tr}\left(\log^2\left(E_{\boldsymbol{\phi}}^{\boldsymbol{\Sigma}}\left(\mathbf{X}\right)\right)\right) \right] \\
&= \underset{\boldsymbol{\phi}, \boldsymbol{\theta}}{\arg\min}\, \mathbb{E}_f\left[ \frac{g^*\left(\mathbf{X}\right)}{f\left(\mathbf{X}\right)}\left( \left(\mathbf{X} - D_{\boldsymbol{\theta}}^{\boldsymbol{\mu}}\left(E_{\boldsymbol{\phi}}^{\boldsymbol{\mu}}\left(\mathbf{X}\right)\right)\right)^{\top}\left(\mathbf{X} - D_{\boldsymbol{\theta}}^{\boldsymbol{\mu}}\left(E_{\boldsymbol{\phi}}^{\boldsymbol{\mu}}\left(\mathbf{X}\right)\right)\right) + \frac{1}{d_z}\mathrm{Tr}\left(\log^2\left(E_{\boldsymbol{\phi}}^{\boldsymbol{\Sigma}}\left(\mathbf{X}\right)\right)\right) \right) \right] \\
&= \underset{\boldsymbol{\phi}, \boldsymbol{\theta}}{\arg\min}\, \mathbb{E}_f\left[ \frac{\widetilde{g}^*\left(\mathbf{X}\right)}{\widetilde{f}\left(\mathbf{X}\right)}\left( \left(\mathbf{X} - D_{\boldsymbol{\theta}}^{\boldsymbol{\mu}}\left(E_{\boldsymbol{\phi}}^{\boldsymbol{\mu}}\left(\mathbf{X}\right)\right)\right)^{\top}\left(\mathbf{X} - D_{\boldsymbol{\theta}}^{\boldsymbol{\mu}}\left(E_{\boldsymbol{\phi}}^{\boldsymbol{\mu}}\left(\mathbf{X}\right)\right)\right) + \frac{1}{d_z}\mathrm{Tr}\left(\log^2\left(E_{\boldsymbol{\phi}}^{\boldsymbol{\Sigma}}\left(\mathbf{X}\right)\right)\right) \right) \right]. \quad (15)
\end{aligned}
$$

Minimising the penalisation term is equivalent then to make each component of the diagonal of $E_{\boldsymbol{\phi}}^{\boldsymbol{\Sigma}}\left(\mathbf{X}\right)$ close to 1 for every point $\mathbf{x}$ of the data set. Since we pre-train the pair encoder/decoder as a classical autoencoder, we decided to add this penalisation term for normalised data to make sure that this part of the network is also pre-trained.

At last, the full proposed training procedure of the VAE with weighted samples is given in Algorithm 1. Note that this procedure can easily be adapted to the classical VAE framework described in Section 3 only by removing the likelihood ratio $\widetilde{g}^*\left(\mathbf{x}\right)\Big/\widetilde{f}\left(\mathbf{x}\right)$.

---

**Algorithm 1** VAE-IS

---

**Require:** $\left(\mathbf{X}^{(n)}\right)_{n \in [\![1,N]\!]} \sim f, K, d_{\boldsymbol{z}}$
 1: Randomly pick $K$ points $\left(\mathbf{X}^{(s(k))}\right)_{k \in [\![1,K]\!]}$ within the dataset according to the procedure described at the beginning of Section 5.2
 2: Train $\mathrm{VP}_{\boldsymbol{\lambda}}$ by minimising equation 14 and get $\boldsymbol{\lambda}^{(0)}$
 3: Train $\left(E_{\boldsymbol{\phi}}, D_{\boldsymbol{\theta}}\right)$ by minimising equation 15 and get $\left(\boldsymbol{\phi}^{(0)}, \boldsymbol{\theta}^{(0)}\right)$
 4: Train the whole VAE $\left(E_{\boldsymbol{\phi}}, D_{\boldsymbol{\theta}}, \mathrm{VP}_{\boldsymbol{\lambda}}\right)$ by maximising equation 13 starting from $\left(\boldsymbol{\phi}^{(0)}, \boldsymbol{\theta}^{(0)}, \boldsymbol{\lambda}^{(0)}\right)$
 5: **return** Trained VAE.

---

**Remark 5.1.** *The objective of (Wang et al., 2019) is the same as ours: learning a target distribution $g^*$ by using observations from an initial distribution $f$ with a VAE. The main principle of their algorithm is the following:*

1. *train a classical VAE as in Section 3 with the available observations from $f$,*

2. *draw new points from the trained VAE and compute the corresponding likelihood ratios,*

3. *boostrap among the new sample with probabilities proportional to the likelihood ratios.*

*They iteratively repeat this procedure and the final sample is theoretically distributed according to a distribution close to $g^*$. Contrary to their work, we propose a one-step procedure to achieve the same goal by modifying the objective function of the VAE. Moreover, it does not seem clear how to have access to the PDF of the resulting approximating distribution of $g^*$ in (Wang et al., 2019) (if at all possible) because they only have access to a bootstrapped sample, whereas it is straightforward with our approach. In addition, we also introduce a new pre-training procedure of the weights of the neural networks in order to prevent posterior collapse. At last, they only test their algorithm on low dimensional (10 and 6) uni-modal target distributions whereas, as developed in Section 6, we test our procedure on more complex test cases. In Section 6, our procedure is compared with the algorithm of Wang et al. (2019), and is more efficient.*

# 6 Application to high-dimensional non-parametric importance sampling

In this section, we first show how to apply the proposed procedure to the classical framework of importance sampling. Then, in order to illustrate the practical interest of the previous efforts, we evaluate numerically the performances of the suggested procedure and we compare them to the performances of some existing IS methods. The code to reproduce the numerical experiments is publicly available at: `https://github.com/Julien6431/Importance-Sampling-VAE.git`.

## 6.1 Importance sampling supported by a VAE

In this paper, we propose to select the IS auxiliary distribution within the distributions parameterised by VAEs. In the IS framework, the target distribution corresponding to $g^*$ of Section 4 is here $g_{\text{opt}}$, which is known up to constant. Moreover, we assume that the initial sampling distribution $f$ is perfectly known. Then, in order to compute the likelihood ratios $w^g$ of the estimator in equation 1, let us explain how to get the corresponding PDF of the resulting distribution $g_{\boldsymbol{\theta}}$. The most naive way to do so is (Wang et al., 2019):

1. draw a sample $\left(\mathbf{X}^{(n)}, \mathbf{Z}^{(n)}\right)_{n\in[\![1,N]\!]}$ according to the joint distribution $g_{\boldsymbol{\theta}}\left(\mathbf{x}, \mathbf{z}\right) = p\left(\mathbf{z}\right) g_{\boldsymbol{\theta}}\left(\mathbf{x}\,|\,\mathbf{z}\right)$ on $\mathcal{X} \times \mathcal{Z}$,

2. estimate the PDF values associated to the sample with:

$$\widehat{g_{\boldsymbol{\theta}}\left(\mathbf{X}^{(n)}\right)} = \frac{1}{N}\sum_{k=1}^{N} g_{\boldsymbol{\theta}}\left(\mathbf{X}^{(n)}\,\Big|\,\mathbf{Z}^{(k)}\right). \tag{16}$$

However, the use of these estimated values of the PDF makes the IS estimator in equation 1 biased since $\widehat{g_{\boldsymbol{\theta}}\left(\mathbf{X}^{(n)}\right)}$ is at the denominator. Moreover, the PDF values $\left(\widehat{g_{\boldsymbol{\theta}}\left(\mathbf{X}^{(n)}\right)}\right)_{n\in[\![1,N]\!]}$ are not independent. Thus, the estimator in equation 1 is no longer a sum of independent random variables and as a result, its convergence is no longer guaranteed.

Hence, in order to keep the convenient properties of the classical IS estimator in equation 1 (unbiasedness and convergence), we adopt the following procedure:

1. approximate the whole marginal distribution $g_{\boldsymbol{\theta}}$ in equation 2 by:

$$g_{\boldsymbol{\theta}}^{M}\left(.\right) = \frac{1}{M}\sum_{m=1}^{M} g_{\boldsymbol{\theta}}\left(.\,\Big|\,\mathbf{Z}^{(m)}\right), \tag{17}$$

   where $M \geq 1$ and where $\left(\mathbf{Z}^{(m)}\right)_{m\in[\![1,M]\!]}$ is an i.i.d. sample from the latent space $\mathcal{Z}$ distributed according to the prior distribution $p$,

2. draw a sample $\left(\mathbf{X}^{(n)}\right)_{n\in[\![1,N]\!]}$ according to $g_{\boldsymbol{\theta}}^{M}$ and compute the estimator $\widehat{I}_{g_{\boldsymbol{\theta}}^{M},N}^{\text{IS}}$.

The main point of this new procedure is that we no longer work with $g_{\boldsymbol{\theta}}$ at all but only with the distribution $g_{\boldsymbol{\theta}}^{M}$ for which we can generate new points and compute exactly and straightforwardly the PDF for all $\mathbf{x} \in \mathcal{X}$. In that case, we come back to the classical and well-known IS framework from Section 2 and $g_{\boldsymbol{\theta}}^{M}$ allows to keep the convenient statistical properties of the IS estimator.

**Proposition 6.1.** *For every fixed $M \geq 1$, the IS estimator $\widehat{I}_{g_{\boldsymbol{\theta}}^{M},N}^{IS}$ is unbiased and convergent as $N \to +\infty$.*

*Proof.* See Appendix A. □

The detailed architecture of the VAE used for the following test cases is shown in Figure 1. Moreover, for each test case, we set $K = 75$ and $M = 10^3$, and we will explicit the dimension $d_z$ of the latent space chosen for each example.

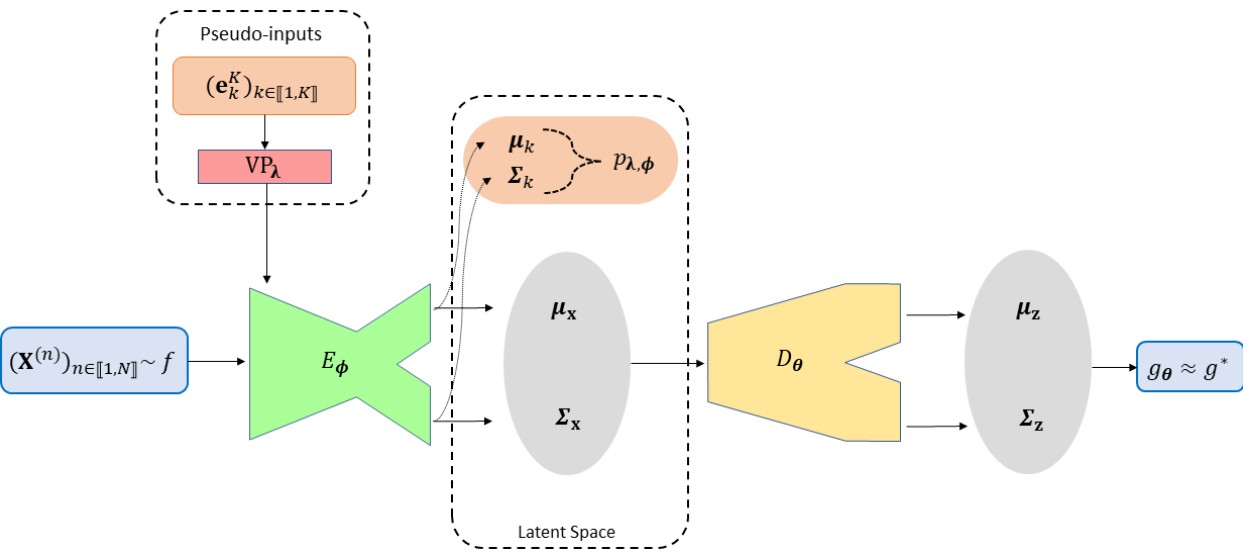

Figure 1: Representation of our suggested VAE architecture for IS in high dimension.

## 6.2 Generation of samples by adaptive IS

A first application of density estimation with weighted samples is adaptive IS for sampling from a target distribution $g^*$. Sometimes, sampling directly from the target distribution $g^*$ is not possible, especially when it is only known up to a constant (Ghosal & van der Vaart, 2017). Then, starting from an initial proposal distribution $f$, adaptive IS algorithms consist in iteratively approximating the target distribution within a family of proposal distributions and at the end, the resulting sample is expected to be drawn according to a distribution close to $g^*$.

In both examples, we use a classical adaptive IS scheme (AIS) as in (Marin et al., 2019), except that we return the weighted sample generated only at the last iteration. We use a distribution parameterised by a VAE as the proposal distribution at each iteration (AIS-VAE). The detailed pseudo-code of the procedure is described in Algorithm 2.

---

**Algorithm 2** AIS-VAE

---

**Require:** $f$, $g^*$ $N$, $K$, $d_{\boldsymbol{z}}$, $M$, $it_{\max}$

1: Draw $\left(\mathbf{X}_n^{(0)}\right)_{n\in[\![1,N]\!]}$ according to $f$

2: Compute $w_n^{(0)} = g^*\left(\mathbf{X}_n^{(0)}\right) / f\left(\mathbf{X}_n^{(0)}\right)$

3: **for** $it \in [\![1, it_{\max}]\!]$ **do**

4:    Train a VAE $(E_{\boldsymbol{\phi}}, D_{\boldsymbol{\theta}}, \mathrm{VP}_{\boldsymbol{\lambda}})$ according to the procedure in Algorithm 1 with the weighted sample $\left(\mathbf{X}_n^{(it-1)}, w_n^{(it-1)}\right)$

5:    Compute the estimate of the resulting distribution $g_{\boldsymbol{\theta},it}^M$

6:    Draw $\left(\mathbf{X}_n^{(it)}\right)_{n\in[\![1,N]\!]}$ according to $g_{\boldsymbol{\theta},it}^M$

7:    Compute $w_n^{(it)} = g^*\left(\mathbf{X}_n^{(it)}\right) / g_{\boldsymbol{\theta},it}^M\left(\mathbf{X}_n^{(it)}\right)$

8: **end for**

9: **return** Weighted sample $\left(\mathbf{X}_n^{(it)}, w_n^{(it)}\right)$

---

### 6.2.1 Example in dimension 10

First, let us consider a 10-dimensional bimodal target distribution given by:

$$g_1^* \propto \mathcal{N}\left(2.5 \times \mathbf{1}_{10}, \mathbf{I}_{10}\right) + \mathcal{N}\left(-2.5 \times \mathbf{1}_{10}, \mathbf{I}_{10}\right), \tag{18}$$

where $\mathbf{1}_{10} = (1, \ldots, 1)^\top \in \mathbb{R}^{10}$ and $\mathbf{I}_{10}$ is the 10-dimensional identity matrix. For comparison purposes, we execute as well the AIS algorithm with a Gaussian mixture with two components (AIS-GM) as the proposal distribution. We also execute the algorithm introduced in Wang et al. (2019) with the proposed setting (AIS-Wang et al. (2019)) as well as the algorithm introduced in this article but with a standard Gaussian prior and without the pre-training procedure of Section 5.2 (AIS-standard VAE). We will also use the Kullback-Leibler divergence (Kullback & Leibler, 1951) to quantitatively compare the performances of every method, since it is a classical measure to quantify the dissimilarity between two distributions. However, its value strongly depends on both compared distributions and also on their dimension. Therefore, it is not possible to have a clear idea of the expected scale and it is only relevant to compare multiple values. The standard Gaussian distribution in dimension 10 is the starting proposal distribution for both algorithms. We perform $n_{\text{rep}} = 100$ executions of each algorithm with 10 iterations. Then, at each iteration, we draw $N = 10^4$ new points, and the dimension of the latent space chosen for this problem is $d_z = 4$.

For every algorithm, the results are as follows: either both modes, and so the whole target distribution, are perfectly found and approximated, or only one mode is found but it is well approximated. Examples of the representation of the final approximating sample are given in Figure 2. However, the main difference between the four algorithms is their success rate, i.e. the frequency with which the resulting distribution finds both modes. In the first place, as shown in Table 1, the AIS-VAE algorithm finds both modes more than 70% of the time and gives a very good approximation of the target distribution. In the second place, the AIS-GM algorithm only finds both modes less than 10% of the time over the $n_{\text{rep}} = 100$ repetitions. Third, the algorithm of Wang et al. (2019) is intermediate because it finds both modes 46% of the time. These results show that a distribution parameterised by a VAE is more likely to catch the characteristics of a target distribution than a single Gaussian or a Gaussian mixture, and even though the latter knew the number of modes in advance. Moreover, the use of a tempered version of both the encoder and the decoder as in Wang et al. (2019) definitely helps to explore a larger area of both the latent space and the input space and thus to catch the characteristics of a target distribution, compared to the AIS procedure with a standard VAE with a standard Gaussian prior and without the pre-training procedure as the IS auxiliary sampling distribution. The latter has a success rate of 25%. Nevertheless, the use of a standard Gaussian prior prevents to reach better performances on a multimodal problem as our full procedure. Thus, these numerical results strongly highlight the interest of the use of both VampPrior and of the pre-training procedure for multimodal problems.

The likelihood ratios corresponding to the samples adaptively guide the algorithm from the initial distribution $f$ to the target one $g_1^*$. However, even more so when the dimension increases, one or some values can be significantly larger than the others, and as a result the adaptive IS algorithm will go in that direction. Nevertheless, we can not explain why this phenomenon seems such less important when using a VAE rather than a Gaussian mixture. A first possibility is that the dimensionality reduction performed by the VAE makes the process more robust. A second one can be the differences between the two learning procedures themselves. The learning procedure is based on mini-batches for the VAE whereas the whole dataset is used in a single step for learning the Gaussian mixture.

Table 1: Comparison of the four algorithms. The first row of the table represents the success rate of the corresponding algorithm and the quantity $(D_{\text{KL}})_{\text{mean}}^{\text{success}}$ represents the mean value of the Kullback-Leibler divergence between the empirical generated distribution and the target $g_1^*$ over the successful samples.

| | AIS-VAE | AIS-GM | AIS-Wang et al. (2019) | AIS-standard VAE |
|---|---|---|---|---|
| Success rate | 72% | 7% | 46% | 25% |
| $(D_{\text{KL}})_{\text{mean}}^{\text{success}}$ | $2.48 \times 10^{-2}$ | $1.13 \times 10^{-1}$ | $4.69 \times 10^{-1}$ | $1.21 \times 10^{-1}$ |

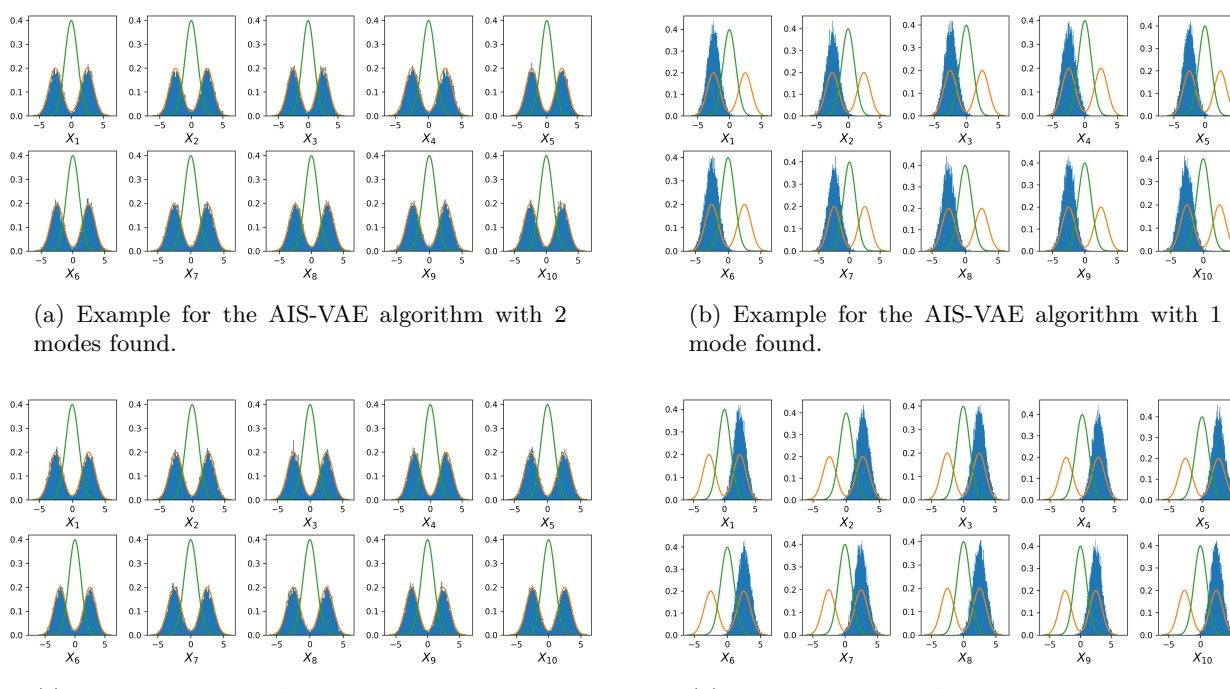

(a) Example for the AIS-VAE algorithm with 2 modes found.

(b) Example for the AIS-VAE algorithm with 1 mode found.

(c) Example for the AIS-GM algorithm with 2 modes found.

(d) Example for the AIS-GM algorithm with 1 mode found.

Figure 2: Graphical representation as 10 histograms of the 10 marginals of the final generated sample for each case for the AIS-VAE and AIS-GM algorithms. The continuous orange line represents the PDF of the marginals of the target distribution $g_1^*$. The continuous green line represents the PDF of the marginals of the starting distribution $f$.

### 6.2.2 Example in dimension 20

Second, let us consider a 20-dimensional target distribution $g_2^*$ defined by the marginal distributions and the dependence structure given in Table 2. The starting distribution chosen here is $f = \mathcal{N}\left(\mathbf{0}_{20}, 2 \times \mathbf{I}_{20}\right)$. We perform 10 iterations and we draw $N = 10^4$ new points at each one. The dimension of the latent space for this problem is $d_z = 8$. Examples of the resulting empirical distribution returned by the AIS-VAE algorithm can be found in Figure 3. Moreover, since the target distribution $g_2^*$ has only one mode, it is interesting to note that the performances of the AIS-standard VAE algorithm (not reported in Figure 3) are exactly the same as the ones of the full proposed procedure in Algorithm 2. We can see graphically that the target distribution is most of the time well approximated by the resulting distribution. However, because of very high negative values of the log-PDF, it is not possible to compute numerically the Kullback-Leibler divergence between the empirical distribution and the target distribution $g_2^*$, and then to use it as a quantitative measure of the quality of the approximation. Indeed, we recall that the expression of the forward Kullback-Leibler divergence between the target distribution $g_2^*$ and the approximating one $\widehat{g}_2$ is $D_{\mathrm{KL}}\left(g_2^* \| g\right) = \mathbb{E}_{g_2^*}\left[\log\left(g_2^*\left(\mathbf{X}\right)\right) - \log\left(\widehat{g}_2\left(\mathbf{X}\right)\right)\right]$. For estimating this quantity, we first draw an i.i.d. sample $\left(\mathbf{X}^{(n)}\right)_{n \in \llbracket 1, N \rrbracket}$ from $g_2^*$, we compute the log-PDF values for each generated point and we return the empirical mean. However, if for some $n \in \llbracket 1, N \rrbracket$ the PDF $\widehat{g}_2\left(\mathbf{X}^{(n)}\right)$ is very very close to 0, the corresponding log-value $\log\left(\widehat{g}_2\left(\mathbf{X}^{(n)}\right)\right)$ will be a very large negative value. Then, it might be possible that the software, Python for us, can not handle a such high value and consider it as "Not a number" (NaN). If so, the estimated Kullback-Leibler divergence will also be a NaN.

Moreover, for comparison purposes, we applied the AIS algorithm with a single Gaussian (AIS-SG) as the proposal distribution, since $g_1^*$ has only one mode. However, it does not work at all with the same setting. Indeed, the starting distribution $f = \mathcal{N}\left(\mathbf{0}_{20}, 2 \times \mathbf{I}_{20}\right)$ is too far away from the target $g_2^*$ for the AIS-SG algorithm. Then, it is necessary to choose $f$ closer to $g_2^*$, by modifying the mean vector for example. Once more, this example shows that using a distribution parameterised by a VAE allows to have a larger flexibility

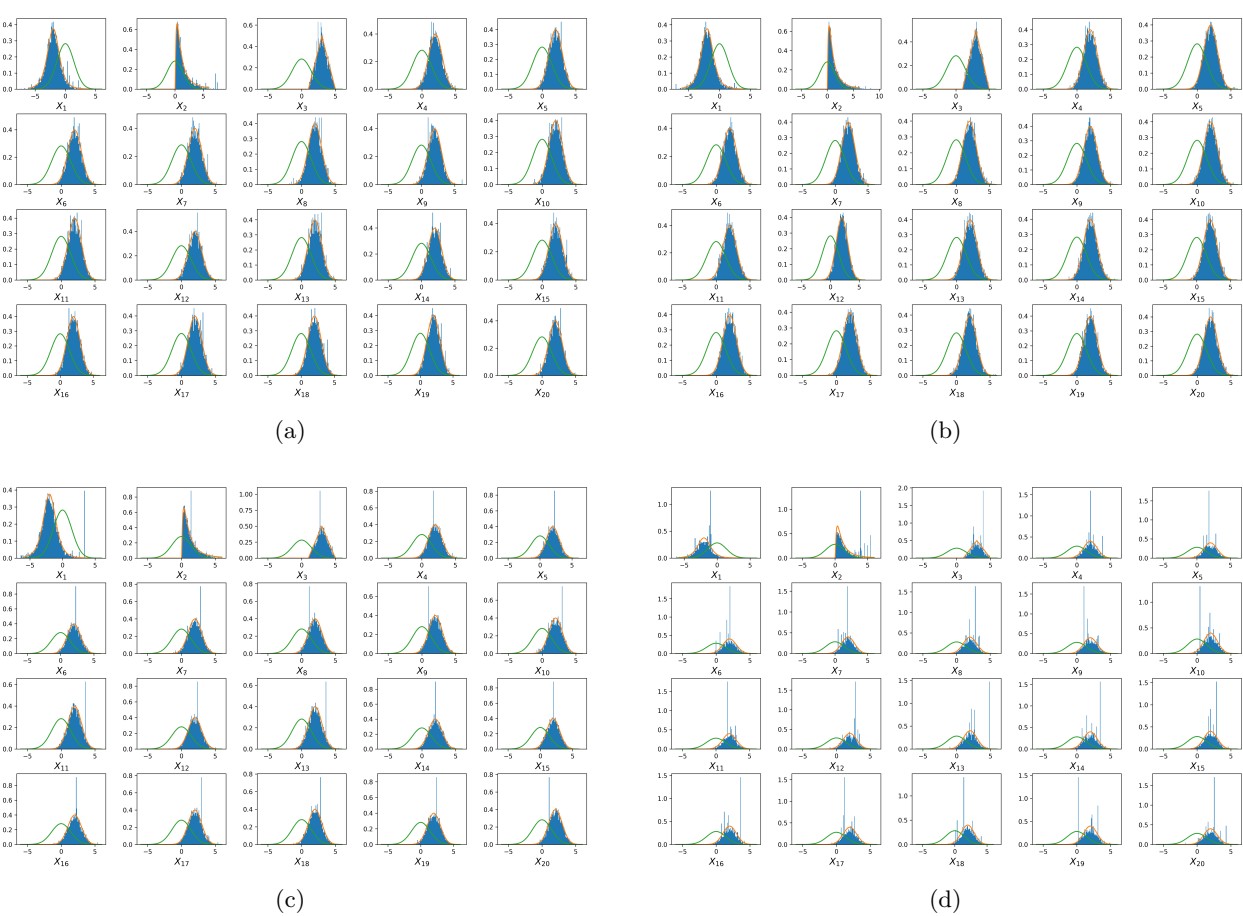

Figure 3: Graphical representation as 20 histograms of the 20 marginals of four realisations of the final generated sample for the AIS-VAE algorithm. The continuous orange line represents the PDF of the marginals of the target distribution $g_2^*$. The continuous green line represents the PDF of the marginals of the starting distribution $f$.

Table 2: Student $(\nu, \mu, \sigma)$: 1-dimensional Student distribution with $\nu > 0$ degrees of freedom, with mean $\mu \in \mathbb{R}$ and scale parameter $\sigma > 0$. LogN $(\mu, \sigma)$: distribution of $\exp(A)$ with $A$ a 1-dimensional Gaussian random variable of mean $\mu \in \mathbb{R}$ and scale $\sigma > 0$. Triangular $(a, m, b)$: 1-dimensional Triangular distribution where $a < b$ are the lower and upper bounds and where $m \in [a, b]$ is the mode. The correlation matrix is given by $R_{i,j} = \mathbf{1}\,(i = j) + 1/4 \times \mathbf{1}\,(|i - j| = 1)$ for all $(i, j) \in [\![1, 20]\!]^2$.

| Input | Distribution |
|---|---|
| $X_1$ | Student $(4, -2, 1)$ |
| $X_2$ | LogN $(0, 1)$ |
| $X_3$ | Triangular $(1, 3, 5)$ |
| $X_4$ to $X_{20}$ | $\mathcal{N}\,(2, 1)$ |
| Copula | Normal copula with correlation matrix $\mathbf{R} \in \mathbb{R}^{20 \times 20}$ |

than with a classical single Gaussian distribution, because the corresponding adaptive IS algorithms have a larger range. At last, we also applied the algorithm proposed in Wang et al. (2019). However, it does not work at all neither. Indeed, as explained in Section 2, at each iteration of an adaptive importance sampling algorithm, we need to compute the likelihood ratios. We recall that the expression to compute the PDF values of the resulting distribution in their algorithm is given in equation 16. Then, it seems that this expression as a sum makes the computation of the PDF values highly numerically unstable when the dimension increases. Some of them are infinite and as a result, an error arises.

Finally, we point out that because of the weight degeneracy phenomenon (Rubinstein & Glynn, 2009) which occurs in IS when the dimension increases, the AIS-VAE algorithm, in the same manner as the other adaptive IS algorithms, will not work very well in higher dimension than 20.

### 6.3 Adaptive importance sampling for estimating failure probabilities

Another application of density estimation with weighted samples is adaptive IS for reliability analysis. Reliability analysis consists in the estimation of the failure probability $p_t = \mathbb{P}\,(\psi\,(\mathbf{X}) > t) = \mathbb{E}_f\,[\mathbf{1}\,(\psi\,(\mathbf{X}) > t)]$ for a fixed known threshold $t \in \mathbb{R}$. Classical Monte Carlo sampling is not adapted to this problem when $p_t$ is getting smaller because its computational cost becomes too large to obtain an accurate estimation. Therefore, several techniques have been developed in order to reduce the variance of the estimation of $p_t$: one can mention FORM/SORM methods (Hasofer & Lind, 1974; Breitung, 1984), subset sampling (Cérou et al., 2012) or line sampling (Koutsourelakis et al., 2004) for example.

In the IS framework, because of the rarity of the failure event, it is challenging to directly pick the best representative of $g_{\text{opt}}\,(\mathbf{x}) \propto \mathbf{1}\,(\psi\,(\mathbf{x}) > t)\,f\,(\mathbf{x})$ within the parametric family of densities by minimising the KL divergence to $g_{\text{opt}}$. To overcome this issue, the authors of (Rubinstein & Kroese, 2004; De Boer et al., 2005) introduced the so-called *multi-level cross entropy* method. At each iteration of the algorithm, the cross-entropy problem is solved for a less rare intermediate failure event until reaching the true one. The intermediate events are set from a quantile parameter $\rho$. Several parametric families of densities have been used for this algorithm in the literature: the Gaussian family (Rubinstein & Kroese, 2004; De Boer et al., 2005), the Gaussian mixture family (Kurtz & Song, 2013; Geyer et al., 2019) and the vMFNM family (Wang & Song, 2016; Papaioannou et al., 2019) for example. For the following two test cases, we use the classical multi-level cross-entropy but with auxiliary densities parameterised by VAEs (CE-VAE) and in both of them, we set the dimension of the latent space to $d_z = 2$. The detailed pseudo-code of the procedure is described in Algorithm 3.

At each iteration of the algorithm, we draw $N_{it} = 10^4$ points and the quantile parameter is set to $\rho = 0.25$ for the first example and to $\rho = 0.15$ for the second. We perform $n_{\text{rep}} = 100$ realisations of each estimator to have an estimation of the error. First, we compare the performances of Algorithm 3 with the cross-entropy algorithm using a VAE with a standard Gaussian prior and without the pre-training procedure of Section 5.2 as the IS auxiliary distribution (CE-standard VAE). Second, since the Gaussian and the Gaussian mixture families do not perform well at all in high dimension, we will compare the CE-VAE algorithm with the multi-level cross-entropy algorithm with mixtures of vMFNM as auxiliary densities (CE-vMFNM). As said

---

**Algorithm 3** Cross-entropy algorithm with a VAE as the auxiliary density

---

**Require:** $f$, $N$, $\rho \in [0,1]$, $K$, $d_{\boldsymbol{z}}$, $M$, $it = 0$

1: Draw $\left( \mathbf{X}_n^{(0)} \right)_{n \in [\![1,N]\!]}$ according to $f$

2: Evaluate $y_n^{(0)} = \psi \left( \mathbf{X}_{(n)}^{(0)} \right)$ for all $n \in [\![1, N]\!]$

3: Sort $y_{(1)}^{(0)} \leq \cdots \leq y_{(N)}^{(0)}$ and set $\gamma_0 = \min \left( y_{\lfloor (1-\rho)N \rfloor}^{(0)}, t \right)$

4: Compute $w_n^{(0)} = \mathbf{1} \left( y_n^{(0)} > \gamma_0 \right)$

5: **while** $\gamma_{it} < t$ **do**

6:    $it = it + 1$

7:    Train a VAE $(E_{\boldsymbol{\phi}}, D_{\boldsymbol{\theta}}, \mathrm{VP}_{\boldsymbol{\lambda}})$ according to the procedure in Algorithm 1 with the weighted sample $\left( \mathbf{X}_n^{(it-1)}, w_n^{(it-1)} \right)$

8:    Compute the estimate of the resulting distribution $g_{\boldsymbol{\theta},it}^M$

9:    Draw $\left( \mathbf{X}_n^{(it)} \right)_{n \in [\![1,N]\!]}$ according to $g_{\boldsymbol{\theta},it}^M$

10:    Evaluate $y_n^{(it)} = \psi \left( \mathbf{X}_n^{(it)} \right)$ for all $n \in [\![1, N]\!]$

11:    Sort $y_{(1)}^{(it)} \leq \cdots \leq y_{(N)}^{(it)}$ and set $\gamma_{it} = \min \left( y_{\lfloor (1-\rho)N \rfloor}^{(it)}, t \right)$

12:    Compute $w_n^{it} = \mathbf{1} \left( y_n^{(it)} > \gamma_{it} \right) f \left( \mathbf{X}_n^{(it)} \right) \Big/ g_{\boldsymbol{\theta},it}^M \left( \mathbf{X}_n^{(it)} \right)$

13: **end while**

14: **return** $N^{-1} \sum_{n=1}^N w_n^{it}$

---

in Section 2.2, the latter algorithm requires the number of components of the mixture, so we will test different setups. For comparison purposes, let us define the following quantities:

- $N_{\mathrm{tot}}$: the number of calls to the function for each execution of the algorithm,

- $\widehat{p}_t^{\mathrm{mean}}$: the mean estimated failure probability over the $n_{\mathrm{rep}}$ realisations,

- $\mathrm{COV}(\widehat{p}_t)$: the coefficient of variation of the estimator over the $n_{\mathrm{rep}}$ realisations,

- $\nu_{\mathrm{MC}}$: a coefficient allowing to compare the efficiency of the method compared to the classical Monte Carlo method defined by $\nu_{\mathrm{MC}} = N_{\mathrm{tot}}^{\mathrm{MC}} / N_{\mathrm{tot}}$, where $N_{\mathrm{tot}}^{\mathrm{MC}} = (1 - p_t) \Big/ \left( p_t \mathrm{COV}(\widehat{p}_t)^2 \right)$ is the size of the required sample to reach the same coefficient of variation as $\mathrm{COV}(\widehat{p}_t)$ with the classical Monte Carlo method. If $\nu_{\mathrm{MC}} > 1$, the method is more efficient than Monte Carlo and conversely.

### 6.3.1 Four branches

First, let us consider the analytical problem (Chiron et al., 2023) given for any even dimension $d \geq 2$ by:

$$\forall \mathbf{x} \in \mathbb{R}^d, \ \psi_1(\mathbf{x}) = \min \left\{ \begin{array}{c} \frac{1}{\sqrt{d}} \sum_{i=1}^d x_i \\ -\frac{1}{\sqrt{d}} \sum_{i=1}^d x_i \\ \frac{1}{\sqrt{d}} \left( \sum_{i=1}^{d/2} x_i - \sum_{i=d/2+1}^d x_i \right) \\ \frac{1}{\sqrt{d}} \left( -\sum_{i=1}^{d/2} x_i + \sum_{i=d/2+1}^d x_i \right) \end{array} \right\}. \tag{19}$$

The input vector associated to this function is the standard Gaussian distribution in dimension $d$. The failure threshold is set to $t = 3.5$ such that the theoretical value of the failure probability is $p_t = 9.3 \times 10^{-4}$. This problem is challenging because it has 4 failure modes. A graphical representation of this function in dimension $d = 2$ as well as the failure limit state are represented in Figure 4a. Here, we set the dimension

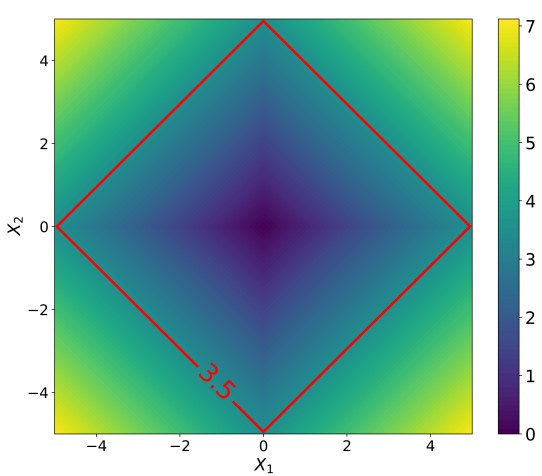
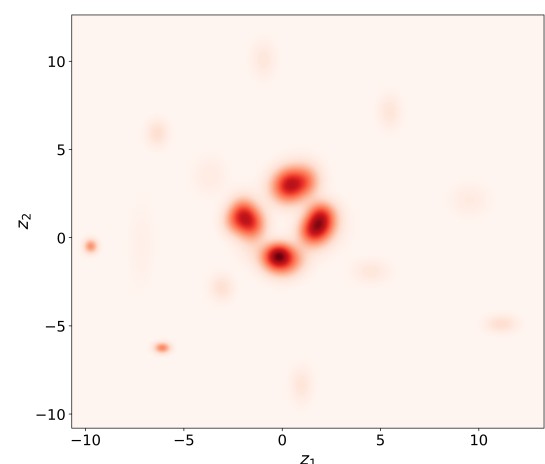

(a) Two-dimensional representation of the function $\psi_1$ as well as the failure limit state corresponding to $t = 3.5$ in red.

(b) Representation of the two-dimensional PDF of the prior distribution in the latent space associated to the last iteration of one execution of the CE-VAE algorithm.

Figure 4

of the problem to $d = 100$, and the numerical performances of the algorithms are given in Table 3. By comparing both the coefficient of variation and the coefficient $\nu_{\mathrm{MC}}$ associated to each estimator, we can see that the proposed CE-VAE algorithm provides better performances than the CE-vMFNM algorithm here, and so without any prior knowledge on the number of failure modes. Figure 4b represents the PDF of the prior distribution in the latent space at the last iteration for one execution of the CE-VAE algorithm. One can clearly see that the four failure modes are well represented in it. This is a clear illustration of the interest of both VampPrior and of the pre-training procedure on a high-dimensional multimodal problem. Therefore when generating new points the four modes will be represented. Moreover, one can note that the choice of the number of components in the mixture for the CE-vMFNM algorithm is crucial. Indeed, since the current problem has 4 failure modes, we need at least 4 components in the mixture in order to get good results. The algorithm for 3 components does not always converge because it sometimes reaches the maximal number of iterations, and the estimation error is large. In addition, unreported visual inspection shows that the empirical distribution of the $n_{\mathrm{rep}} = 100$ realisations of the CE-vMFNM4 algorithm has a heavy tail, and therefore explains the observed error. At last, the results of the CE-standard VAE algorithm are a clear illustration of the huge negative impact of the posterior collapse phenomenon on the performances of the algorithm. Indeed, without VampPrior and the pre-training procedure, the standard VAE never finds any of the four modes and as a result, the corresponding algorithm never converges since it always reaches the maximal number of iterations.

Table 3: Comparison of the CE-VAE algorithm with the CE-vMNFM algorithm with different numbers of components and the CE-standard VAE algorithm for the four branches problem. The number after the "CE-vMFNM" acronym represents the number of components of the mixture given as an algorithm input.

| | CE-VAE | CE-vMFNM3 | CE-vMFNM4 | CE-vMFNM5 | CE-standard VAE |
|---|---|---|---|---|---|
| $N_{\mathrm{tot}}$ | 40000 | 88000 | 50000 | 50000 | 200000 |
| $\widehat{p}_t^{\mathrm{mean}}$ | $9.310 \times 10^{-4}$ | $1.319 \times 10^{-3}$ | $9.835 \times 10^{-4}$ | $9.315 \times 10^{-4}$ | $9.446 \times 10^{-4}$ |
| COV $(\widehat{p}_t)$ | 5.31% | 512.8% | 31.3% | 7.56% | 34.83% |
| $\nu_{\mathrm{MC}}$ | 9.54 | $4.64 \times 10^{-4}$ | $2.19 \times 10^{-1}$ | 3.76 | $4.43 \times 10^{-2}$ |

### 6.3.2 Duffing oscillator

The second example is the Duffing oscillator introduced in (Zuev, 2009), and we consider here its writing discretised in the frequency domain as in (Shinozuka & Deodatis, 1991; Papaioannou et al., 2019). The function of interest is the maximal displacement of the oscillator at $a_{\max} = 2$sec given by:

$$\psi_2 (\mathbf{x}) = \min \{u_1 - u (a_{\max}), u (a_{\max}) - u_2\}, \tag{20}$$

where $u_1 = 0.1$ and $u_2 = -0.06$, and where the displacement $u(a)$ of the oscillator satisfies for all $a \geq 0$:

$$m\ddot{u}(a) + c\dot{u}(a) + k \left(u(a) + \gamma u(a)^3\right) = -m\sigma \sum_{i=1}^{d/2} \left(X_i \cos(\omega_i a) + X_{d/2+i} \sin(\omega_i a)\right), \tag{21}$$

with $m = 1000$ kg, $c = 200\pi$ Ns/m, $k = 1000 (2\pi)^2$ N/m, $\gamma = 1$ m$^{-2}$, $w_i = i\Delta\omega$, $\Delta\omega = 30\pi/d$ and $\sigma = \sqrt{0.01\Delta\omega}$. Moreover, the initial conditions are set to $u(0) = 0$ and $\dot{u}(0) = 1.5$, the random variable $\mathbf{X}$ follows a standard Gaussian distribution of dimension $d = 200$ and the failure threshold is set to $t = 0$. It is possible to verify that there are then two failure modes. The reference value of the failure probability is $p_{\mathrm{ref}} = 4.28 \times 10^{-4}$ and it has been computed with a large Monte Carlo estimation of size $10^6$ (Chiron et al., 2023).

Table 4: Comparison of the CE-VAE algorithm with the CE-vMNFM algorithm with different number of components and the CE-standard VAE algorithm on the Duffing oscillator problem. The number after the "CE-vMFNM" acronym represents the number of components of the mixture given as an algorithm input.

| | CE-VAE | CE-vMFNM1 | CE-vMFNM2 | CE-vMFNM3 | CE-standard VAE |
|---|---|---|---|---|---|
| $N_{\mathrm{tot}}$ | 30000 | 200000 | 40000 | 40000 | 47900 |
| $\widehat{p}_t^{\mathrm{mean}}$ | $4.27 \times 10^{-4}$ | $4.09 \times 10^{-4}$ | $4.25 \times 10^{-4}$ | $4.26 \times 10^{-4}$ | $3.39 \times 10^{-4}$ |
| COV $(\widehat{p}_t)$ | 8.69% | 45.85% | 3.72% | 2.51% | 31.09% |
| $\nu_{\mathrm{MC}}$ | 10.3 | $5.55 \times 10^{-2}$ | 42.2 | 92.8 | $5.04 \times 10^{-1}$ |

The numerical performances of the algorithms are given in Table 4. By comparing once more both the coefficient of variation and the coefficient $\nu_{\mathrm{MC}}$ associated to each estimator, the CE-VAE algorithm provides satisfying performances, but the CE-vMFNM2 and CE-vMFNM3 algorithms outperform it. Indeed, their coefficient of variation is between two and three times smaller than the one of the CE-VAE algorithm and although the CE-vMFNM algorithms require one more iteration to converge, their associated coefficient $\nu_{\mathrm{MC}}$ is much better. The current problem is probably very well suited for this family of auxiliary distributions and can explain these very good results in such high dimension. However, as already said in the previous example, contrary to the proposed CE-VAE algorithm, the CE-vMFNM algorithm requires the knowledge of the number of components in the mixture, which can often be a limitation. Here, at least two components are required because we can observe that the CE-vMFNM1 algorithm does not work at all since it reaches the maximal number of iteration for each execution. Then, the proposed training procedure of the VAE allows to identify both failure modes without any prior knowledge and to get satisfying results anyway. At last, in the same way as the previous example, the CE-standard VAE algorithm often suffers from the posterior collapse phenomenon and does not perform as well as the CE-VAE algorithm.

## 7 Conclusion

In the present article, we are interested in probability density estimation with weighted samples, i.e. the estimation of a target distribution $g^*$ by using a sample drawn from another distribution $f$. It is a major topic of interest in statistics which has many applications, some of them presented in Section 6. We suggest to approach $g^*$ by a distribution parameterised by a VAE. To do so, we extend the existing VAE framework to the case of weighted samples, by introducing the new objective function wELBO to maximise in order to learn the best parameters of the neural networks. Even if the corresponding distribution theoretically belongs to a parametric family, its characteristics make it closer to a non-parametric model. Despite the

very high number of parameters to estimate, this family is much more efficient in high dimension than the classical Gaussian or Gaussian mixture families. Moreover, in order to add flexibility to the model, and more precisely to be able to learn multimodal distributions, we use a learnable prior distribution for the latent variable called VampPrior. We also introduce a new pre-training procedure for the VAE in order to find good starting points $\left(\boldsymbol{\phi}^{(0)}, \boldsymbol{\theta}^{(0)}, \boldsymbol{\lambda}^{(0)}\right)$ for the maximisation of wELBO and to prevent as much as possible the posterior collapse phenomenon to happen. At last, we illustrate and discuss the practical interest of the proposed method. We first describe the classical IS framework for estimating an expectation and we show how to exploit the resulting distribution to that purpose. Then, we apply the proposed procedure in an adaptive IS algorithm for drawing points according to a target distribution. Both examples in dimension 10 and 20 show that a VAE has a larger flexibility and is more likely to catch the specificity of a distribution than a Gaussian or a Gaussian mixture distribution. Finally, we introduce the proposed method in an adaptive IS algorithm for estimating a failure probability in high dimension. We observe that the resulting estimation is always quite accurate without any prior knowledge on the form of the problem, in opposition to the vMNFM family. These numerical examples also strongly illustrate and support the use of VampPrior and of the pre-training procedure for multimodal problems.

It is important to note that the computational time of the training of neural networks, and so of a VAE, is growing with the number of training points and the input dimension. As a comparison, the training time of the other parametric families used in this article, not relying on neural networks, is almost instantaneous whereas one execution of the proposed procedure can last from less than one minute to 5 minutes on a CPU. However, once the VAE is trained, the generation of new points is instantaneous, and when considering time-expensive black-box functions, the training time of a VAE becomes negligible. Moreover, the suggested architecture in Figure 1 is not the only possibility. Indeed, we suggest here a unique architecture quite simple to implement but general enough to be adapted for every considered problem. However, if it is necessary, an interested user can modify the suggested architecture to be more adapted to its data.

Finally, this article shows that a VAE is a powerful tool for density estimation with or without weighted samples, and we apply it for adaptive IS. Then, one can investigate the use of VAEs for others algorithms or uncertainty quantification problems requiring high-dimensional density estimation, such as an improved adaptive IS algorithm for failure probability estimation (Papaioannou et al., 2019) or some MCMC methods (Chib & Greenberg, 1995; Au & Beck, 2001) for example.

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

# A Proof of Proposition 6.1

Let $M \geq 1$ and let $\mathcal{F}_M$ be the $\sigma$-algebra generated by the i.i.d. random variables $\mathbf{Z}^{(1)}, \ldots, \mathbf{Z}^{(M)}$ distributed according to the prior $p$. We recall that $g_{\boldsymbol{\theta}}^M (.) = M^{-1} \sum_{m=1}^{M} g_{\boldsymbol{\theta}} \left( . \, \middle| \mathbf{Z}^{(m)} \right)$ and that the expression of $\widehat{I}_{g,N}^{\mathrm{IS}}$ is given in equation 1.

## A.1 Unbiasedness

We have:

$$
\begin{aligned}
\mathbb{E}\left[\widehat{I}_{g_{\boldsymbol{\theta}}^M,N}^{\mathrm{IS}}\right] &= \mathbb{E}_p\left[\mathbb{E}_{g_{\boldsymbol{\theta}}^M}\left(\widehat{I}_{g_{\boldsymbol{\theta}}^M,N}^{\mathrm{IS}}\,\middle|\,\mathcal{F}_M\right)\right]\\
&= \mathbb{E}_p\left[\mathbb{E}_{g_{\boldsymbol{\theta}}^M}\left(\frac{1}{N}\sum_{n=1}^{N}\psi\left(\mathbf{X}^{(n)}\right)\frac{f\left(\mathbf{X}^{(n)}\right)}{g_{\boldsymbol{\theta}}^M\left(\mathbf{X}^{(n)}\right)}\,\middle|\,\mathcal{F}_M\right)\right]\\
&= \mathbb{E}_p\left[\frac{1}{N}\sum_{n=1}^{N}\mathbb{E}_{g_{\boldsymbol{\theta}}^M}\left(\psi\left(\mathbf{X}^{(n)}\right)\frac{f\left(\mathbf{X}^{(n)}\right)}{g_{\boldsymbol{\theta}}^M\left(\mathbf{X}^{(n)}\right)}\,\middle|\,\mathcal{F}_M\right)\right] \text{ by linearity of the conditional expectation}\\
&= \mathbb{E}_p\left[\mathbb{E}_{g_{\boldsymbol{\theta}}^M}\left(\psi\left(\mathbf{X}\right)\frac{f\left(\mathbf{X}\right)}{g_{\boldsymbol{\theta}}^M\left(\mathbf{X}\right)}\,\middle|\,\mathcal{F}_M\right)\right]\\
&\qquad\qquad \text{ conditionally to } \mathcal{F}_M, \left(\mathbf{X}^{(n)}\right)_{n\in[\![1,N]\!]} \text{ is an i.i.d. sample distributed according to } g_{\boldsymbol{\theta}}^M\\
&= \mathbb{E}_p\left[\mathbb{E}_f\left(\psi\left(\mathbf{X}\right)\middle|\,\mathcal{F}_M\right)\right] \text{ by importance sampling}\\
&= \mathbb{E}_p\left[I\right]\\
&= I.
\end{aligned}
$$

Then, $\widehat{I}_{g_{\boldsymbol{\theta}}^M,N}^{\mathrm{IS}}$ is an unbiased estimator of $I$.

## A.2 Convergence

By the law of large numbers, conditionally to $\mathcal{F}_M$, we have:

$$\widehat{I}^{\mathrm{IS}}_{g_{\boldsymbol{\theta}}^M,N} = \frac{1}{N} \sum_{n=1}^{N} \psi\left(\mathbf{X}^{(n)}\right) \frac{f\left(\mathbf{X}^{(n)}\right)}{g_{\boldsymbol{\theta}}^M\left(\mathbf{X}^{(n)}\right)} \xrightarrow[N\longrightarrow+\infty]{a.s.} I. \tag{22}$$

Then,

$$
\begin{aligned}
\mathbb{P}\left(\widehat{I}^{\mathrm{IS}}_{g_{\boldsymbol{\theta}}^M,N} \xrightarrow[N\to+\infty]{} I\right) &= \mathbb{E}\left[\mathbf{1}\left(\widehat{I}^{\mathrm{IS}}_{g_{\boldsymbol{\theta}}^M,N} \xrightarrow[N\to+\infty]{} I\right)\right] \\
&= \mathbb{E}_p\left[\mathbb{E}_{g_{\boldsymbol{\theta}}^M}\left(\mathbf{1}\left(\widehat{I}^{\mathrm{IS}}_{g_{\boldsymbol{\theta}}^M,N} \xrightarrow[N\to+\infty]{} I\right)\middle|\mathcal{F}_M\right)\right] \\
&= \mathbb{E}_p\left[\underbrace{\mathbb{P}\left(\widehat{I}^{\mathrm{IS}}_{g_{\boldsymbol{\theta}}^M,N} \xrightarrow[N\to+\infty]{} I\middle|\mathcal{F}_M\right)}_{=1}\right] \\
&= \mathbb{E}_p\left[1\right] \\
&= 1.
\end{aligned}
$$

So the estimator $\widehat{I}^{\mathrm{IS}}_{g_{\boldsymbol{\theta}}^M,N}$ converges almost surely to $I$.

