# OpenReview forum: "Variational autoencoder with weighted samples for high-dimensional non-parametric adaptive importance sampling"
_TMLR — Accepted by TMLR_

### Review · Reviewer_89qk · 2023-11-04

**Summary Of Contributions:**

The paper begins by giving a thorough background on VAEs, as well as the Variational Mixture of Posteriors (VampPrior) in Section 2.

Section 3 introduces the methodology and the problem set up. The problem setup is to estimate the normalising constant of an empirical distribution using only samples from another empirical distribution (which is also known up to a constant). The authors propose training a VAE with VampPrior using the importance sampling trick to tackle this problem.

The authors propose a pre-training setup to avoid the problem of posterior collapse. They pre-train the VampPrior, initialising its outputs to be close to points with large ratio g/f.

The authors use their method to provide improved importance samplers. They test their improved importance samples on several tasks in statistics.

**Audience:**

Yes

**Broader Impact Concerns:**

None.

**Claims And Evidence:**

Yes

**Requested Changes:**

problem setup. For example, why is PDF estimation relevant vs. sampling from the distribution? It’s stated briefly in the introduction that the authors apply their approach to important sampling, but I think it would be helpful to continually remind the reader of the problem the authors are trying to solve.

It would be helpful to motivate why it is important to sample points according to the ratio g/f when pre-training the VampPrior.

In autoencoder pre-training (eq. 14), can you further explain the loss term for the diagonal covariance matrix? For example, I think 1) It would be helpful to remark that the output of the encoder/decoder is the log variance and 2) that the encoder/decoder only output d_z entries corresponding to the diagonal of the covariance matrix.

Algorithm 1 should explain how the k points are sampled to pre-train the VampPrior.

Potential typo in the paragraph after equation 15 when stating optimality conditions for g_opt. Should be \forall x \in \mathcal{X}.

In Figure 1, I think it’s unnecessary (i.e. crosses a level of abstraction we don’t need to) to show the Encoder and Decoder are parameterised as MLPs.

I found Section 5.1.2 confusing. It would be helpful to expand this section and provide more exposition.

In Section 5.2.2, can you explain why large values of the PDF prevent computing the KL divergence?

**Strengths And Weaknesses:**

The paper does a good job of providing a thorough background with clear explanation on VAEs.

It’s nice that the paper is structured to provide pointers to related work throughout, rather than having a dedicated section to Related Work.

I think this paper could use some more motivation to help explain the problem setup (see requested changes).

I think the authors should spend more time explaining why PDF estimation is an important problem. The authors state in Remark 4.1 that Wang 2019 does not provide a method to compute the PDF of the desired distribution g^*, but it is unclear to the reader why this is important. The authors should remind the reader that they wish to apply their method to the problem of importance sampling.

The results in section 5.2.1 are interesting, but I find it a weakness of the paper that the phenomenon is not thoroughly understood. I also think there should be some ablations demonstrating which part of the authors approach is necessary to make their method work (e.g. how important is the VampPrior). It would be interesting to see for example if the success of the authors method can be predicted by the result of the pre-training initialisation of the VampPrior and/or the encoder and decoder. An experiment in 2 or 3 dimensions would be helpful, perhaps the phenomenon can be fully explained there.

I also think there should be further explanation for the KL divergence metric used in experiment 5.2.1. This is the only metric that measures the sample quality from the joint distribution. It would be nice to provide some context/interpretation for the metric here, since right now it is hard to interpret its scale (the only takeaway I have right now is that it is lower, hence better, for the authors method compared to the baseline).

---

> ### Author Response · Authors · 2023-11-30
> **Response to Reviewer 89qk part 1**
>
> Dear Reviewer, we are pleased to submit a new version of our paper. We were able to address all your comments. Overall, this leads to a significant improvement of the paper, and we hope that you will be satisfied with this new version. We answer below to your remarks and we comment the modifications made. Of course, we are at your disposal should you require further information.
>
> > problem setup. For example, why is PDF estimation relevant vs. sampling from the distribution? It’s stated briefly in the introduction that the authors apply their approach to important sampling, but I think it would be helpful to continually remind the reader of the problem the authors are trying to solve.
>
> We agree that additional motivations about the problem of PDF estimation are required. As we briefly say in the introduction and now massively explain in the new Section 2, we want to apply the proposed algorithm to a context of importance sampling. At each iteration of an IS algorithm, we need to have access to the PDF of the resulting distribution to be able to compute the likelihood ratios f/g and to perform the next iteration, and not only to sample from the resulting distribution. Moreover, for the specific case of the estimation of an expectation, the IS estimator $\widehat{I}_{g,N}^{\text{IS}}$ explicitly depends on the likelihood ratio and then requires its value. In the revision, we add and expand this motivation in a new background section (Section 2) as suggested.
>
> > It would be helpful to motivate why it is important to sample points according to the ratio g/f when pre-training the VampPrior.
>
> The proposed sampling strategy of the sub-sample of size $K$ with probabilities proportional to $g/f$ for the pre-training of the parameters $\boldsymbol{\lambda}$ ensures that it is distributed according to the target distribution $g^*$ rather than the initial distribution $f$. Therefore, once pre-trained, the initial pseudo-inputs $\boldsymbol{u}\_k^{(0)} = \text{VP}\_{\boldsymbol{\lambda^{(0)}}}\left(\mathbf{e}\_k\right)$ will already be representative of the target distribution $g^*$. As a result, the characteristics of the target distribution $g^*$, such as its modes, will be already caught by $\text{VP}\_{\boldsymbol{\lambda^{(0)}}}$. We have added this additional explanation in Section 5.2 of the revised article.
>
> > In autoencoder pre-training (eq. 14), can you further explain the loss term for the diagonal covariance matrix? For example, I think 1) It would be helpful to remark that the output of the encoder/decoder is the log variance and 2) that the encoder/decoder only output $d_z$ entries corresponding to the diagonal of the covariance matrix.
>
> First, it is true that in practice, $E_{\boldsymbol{\phi}}^{\boldsymbol{\Sigma}}\left(\mathbf{x}\right)$ and $D\_{\boldsymbol{\theta}}^{\boldsymbol{\Sigma}}\left(\mathbf{z}\right)$ are vectors of size respectively $d\_z$ and $d$ representing the diagonals of the output log-covariance matrices. However, we find it more convenient for exposition to write $E\_{\boldsymbol{\phi}}^{\boldsymbol{\Sigma}}\left(\mathbf{x}\right)$ and $D\_{\boldsymbol{\theta}}^{\boldsymbol{\Sigma}}\left(\mathbf{z}\right)$ as square matrices (although we agree that in practice we do not need to store the off-diagonal zeros). Then, minimising the penalisation term $\frac{1}{d\_z}\mathrm{Tr}\left(\log^2\left(E\_{\boldsymbol{\phi}}^{\boldsymbol{\Sigma}}\left(\mathbf{X}\right)\right)\right)$ is equivalent to make each component of the vector $E\_{\boldsymbol{\phi}}^{\boldsymbol{\Sigma}}\left(\mathbf{X}\right)$ close to $1$ for every point $\mathbf{x}$ of the data set. Since we pre-train the pair encoder/decoder as a classical autoencoder, we decided to add this penalisation term for normalised data to make sure that this part of the network is also correctly pre-trained. We expend the meaning of the penalisation term in Section 5.2 of the revised article as well.
>
> > Algorithm 1 should explain how the k points are sampled to pre-train the VampPrior. In Figure 1, I think it’s unnecessary (i.e. crosses a level of abstraction we don’t need to) to show the Encoder and Decoder are parameterised as MLPs.
>
> We agree with these remarks and we made the corresponding modifications.

---

> > ### Author Response · Authors · 2023-11-30
> > **Response to Reviewer 89qk part 2**
> >
> > > I found Section 5.1.2 confusing. It would be helpful to expand this section and provide more exposition.
> >
> > As motivated earlier, it is crucial for our method to have access to the PDF of the built distribution parameterised by a VAE, and it is not straightforward. A naive way to do so have been proposed in (Wang et al, 2019) and consists in estimating the PDF value of the resulting distribution $g\_{\boldsymbol{\theta}}$ with the procedure described at the beginning of the new Section 6.1. However, when performing the estimation of an expectation, this method leads to a biased estimator $\widehat{I}\_{g\_{\boldsymbol{\theta}},N}^{\text{IS}}$ since the estimated values of the PDF appear in the denominator. To overcome this issue, we propose no longer to estimate the PDF values of $g\_{\boldsymbol{\theta}}$ but to replace the whole distribution $g\_{\boldsymbol{\theta}}$ by an estimate $g\_{\boldsymbol{\theta}}^M$ for which we can generate new points and compute exactly and straightforwardly the PDF. Indeed, by construction of the VAE, we have:\begin{equation}
> >     g\_{\boldsymbol{\theta}}\left(.\right) = \int\_{\mathcal{Z}}g_{\boldsymbol{\theta}}\left(.|\mathbf{z}\right)p\left(\mathbf{z}\right)d\mathbf{z}.
> > \end{equation} Then, based on this expression, we introduce an estimate of the whole distribution $g\_{\boldsymbol{\theta}}$ given by the mixture of distributions: \begin{equation}
> >     g\_{\boldsymbol{\theta}}^M\left(.\right) = \dfrac{1}{M}\sum\_{m=1}^Mg_{\boldsymbol{\theta}}\left(.\left|\mathbf{Z}^{(m)}\right.\right),
> > \end{equation} with $M\geq 1$ and $\left(\mathbf{Z}^{(m)}\right)\_{m\in[[1,M]\]}\in\mathcal{Z}^M$ an i.i.d. sample distributed according to the prior $p$. The main point of this new procedure is that we no longer work with $g\_{\boldsymbol{\theta}}$ at all but only with the distribution $g\_{\boldsymbol{\theta}}^M$. Indeed, the IS estimator used in the following sections is: \begin{equation}
> >     \widehat{I}\_{g\_{\boldsymbol{\theta}}^M,N}^{\text{IS}} = \dfrac{1}{N}\sum\_{n=1}^N\psi\left(\mathbf{X}^{(n)}\right)\dfrac{f\left(\mathbf{X}^{(n)}\right)}{g\_{\boldsymbol{\theta}}^M\left(\mathbf{X}^{(n)}\right)},
> > \end{equation} with $\left(\mathbf{X}^{(n)}\right)\_{n\in[[1,N]]}\in\mathcal{X}^N$ an i.i.d. sample distributed according to $g\_{\boldsymbol{\theta}}^M$. At last, we can verify that $\widehat{I}\_{g\_{\boldsymbol{\theta}}^M,N}^{\text{IS}}$ is an unbiased and convergent estimator of $I$. A detailed proof has been added as an appendix in the article.
> >
> > > In Section 5.2.2, can you explain why large values of the PDF prevent computing the KL divergence?
> >
> > We recall that the expression of the Kullback-Leibler divergence between two distributions $f$ and $g$ is $D\_{\text{KL}}\left(f\Vert g\right) = \mathbb{E}\_f\left[\log\left(f\left(\mathbf{X}\right)\right) - \log\left(g\left(\mathbf{X}\right)\right)\right]$. For estimating this quantity, we first draw an i.i.d. sample $\left(\mathbf{X}^{(n)}\right)\_{n\in[[1,N]]}$ from $f$, we compute the log-PDF values for each generated point and we return the empirical mean. However, if for some $n\in[[1,N]]$ the PDF $g\left(\mathbf{X}^{(n)}\right)$ is very very close to $0$, the corresponding log-value $\log\left(g\left(\mathbf{X}^{(n)}\right)\right)$ will be a very large negative value. Then, it might be possible that the software, Python for us, can not handle a such high value and consider it as "Not a number" (NaN). If so, the estimated Kullback-Leibler divergence will also be a NaN. To overcome this issue, we will investigate the use of other measures to quantify the dissimilarity between the target distribution and the estimated one, such as the kernel distance [1] for example.
> >
> > Reference:
> >
> > [1] Jeff M Philips and Suresh Venkatasubramanian. A gentle introduction to the kernel distance. arXiv preprint arXiv:1103.1625,2011.

---

> > > ### Author Response · Authors · 2023-11-30
> > > **Response to Reviewer 89qk part 3**
> > >
> > > > The results in section 5.2.1 are interesting, but I find it a weakness of the paper that the phenomenon is not thoroughly understood. I also think there should be some ablations demonstrating which part of the authors approach is necessary to make their method work (e.g. how important is the VampPrior). It would be interesting to see for example if the success of the authors method can be predicted by the result of the pre-training initialisation of the VampPrior and/or the encoder and decoder. An experiment in 2 or 3 dimensions would be helpful, perhaps the phenomenon can be fully explained there.
> > >
> > > When developing the proposed algorithm for adaptive IS with a VAE, we wanted the method to be general enough to be adapted to all kinds of problems, such as multimodal problems for example. To do so, we first tried to apply a classical VAE with a standard Normal prior and without pre-training on a two-modal problem. However, with that setting, the VAE did not caught both modes and we experienced posterior collapse. Moreover, we also applied the procedure using VampPrior but without any pre-training procedure, but we faced the same problem. We also tried to use $\beta$-VAEs but results were not satisfying. Therefore, we decided to try a new approach by combining both VampPrior and the proposed pre-training procedure and results were much better. A first illustration of the interest of the use of VampPrior and of the pre-training procedure is provided in Figure 4b. One can clearly see in the latent space that the four modes of the multimodal problem have been well caught by the VAE thanks to the whole procedure. In addition, we will motivate and provide empirical evidence of the interest of the procedure on a new multimodal example and additional experiments. We will show on it that without both VampPrior and the pre-training procedure, results are not as good as with both improvements. At last, a hint to predict the success of the method after the pre-training procedure is the number of modes of the pre-trained prior distribution. If it has the same number of modes as the target distribution, the method is very likely to be a success. However, we need to know how many modes the target distribution has to do that in practice.
> > >
> > > > I also think there should be further explanation for the KL divergence metric used in experiment 5.2.1. This is the only metric that measures the sample quality from the joint distribution. It would be nice to provide some context/interpretation for the metric here, since right now it is hard to interpret its scale (the only takeaway I have right now is that it is lower, hence better, for the authors method compared to the baseline).
> > >
> > > The Kullback-Leibler divergence is a classical measure to quantify the dissimilarity between two distributions. However, its value strongly depends on both compared distributions and also on their dimension. Therefore, it is not possible to have a clear idea of the expected scale and it is only relevant to compare multiple values. The same problem arises for every measure quantifying the dissimilarity between two distributions, in our opinion. We added a sentence in Section 6.2 to make this point clearer.

---

### Review · Reviewer_VqJg · 2023-11-16

**Summary Of Contributions:**

In this work, the authors propose a novel algorithm for building a proposal to do importance sampling (IS), and its adaptive counterpart adaptive importance sampling (AIS). Specifically, the proposal is parameterized by a deep generative model trained using an extension to the VAE framework. Specifically, a novel loss function is proposed that allows a deep generative model to be trained using data from a different distribution.

**Audience:**

Yes

**Claims And Evidence:**

No

**Requested Changes:**

Below, I list my requested changes

1. Include background info on IS and AIS in the appropriate background section. **Critical**
2. An explicit related works section. **Critical**
3. Fix citations. **Minor**
4. Address the loss function concern I raised above **Critical**
5. Provide empirical evidence of the usefulness of pre-initialization and posterior collapse **Critical**
6. Rewrite the biased estimator claim in section 5.1.2 **Critical**
7. Revamp results section with comparisons **Critical**

**Strengths And Weaknesses:**

# Strengths

I think the proposed loss function is **very** nifty. Specifically, the prospect of training a generative model using data from a different distribution is very cool and I think could lead to a lot of cool applications.


# Weaknesses

While I like the proposed approach, the paper has a lot of weaknesses. First, I think the writing can be greatly improved. As the novel loss function relies on the importance sampling trick **and** many of the results presented are a combination of the VAE + AIS, I was surprised that the background section didn't go over IS or AIS. I think this would greatly improve the flow of the paper. Also, there was virtually no related works section. I found this very surprising given that the combination of VAEs and IS-based methods is not new, i.e., [1, 2]. Moreover, there are a slew of methods that try to update the proposal of AIS by minimizing a loss function [3, 4]. The related works section would only strengthen the paper it would help highlight the main contribution which is that IS is done over the data used to train the VAE **and not the latents** which is what has been done in previous works. Also, the formatting instructions for TMLR state to use `\citep` and `\citet` but both are missing in the paper. While this is a minor point, it does reduce the readability of the paper.

Next, the weighted ELBO in equation 12 is **not** a lower bound of the expected log marginal likelihood, $E_{g^*}[\log g_\theta(x)]$ nor does it lead to the same gradient as the lower bound in eq 11. Although the normalization constants are not a function of the parameters, it does scale the ELBO and need to be taken into account to ensure that you're optimizing a lower-bound to the expected log marginal likelihood; if the normalization constants were instead an additive term then you could throw them away and would still be optimizing the expected log marginal likelihood. Instead, the proposed lower-bound (and the computed gradients) is a lower-bound to a scaled expected log marginal likelihood, $\frac{c_f^{-1}}{c_g^{-1}} E_{g^*}[\log g_\theta(x)]$. Below, I demonstrate this (where $\mathcal{L}(x)$ is the ELBO, $g^*(x) = \tilde{g}^*(x) c_g^{-1}$, and, $f(x) = \tilde{f}(x) c_f^{-1}$)
$$ \log g_\theta(x) \geq \mathcal{L}(x) $$
$$ \frac{\tilde{g}^*(x)}{\tilde{f}(x)} \log g_\theta(x) \geq \frac{\tilde{g}^*(x)}{\tilde{f}(x)} \mathcal{L}(x) $$
$$ E_f \left[ \frac{\tilde{g}^*(x)}{\tilde{f}(x)} \log g_\theta(x) \right] \geq E_f \left[ \frac{\tilde{g}^*(x)}{\tilde{f}(x)} \mathcal{L}(x) \right] $$
$$ \int f(x) \frac{\tilde{g}^*(x)}{\tilde{f}(x)} \log g_\theta(x) dx \geq \int f(x) \frac{\tilde{g}^*(x)}{\tilde{f}(x)} \mathcal{L}(x) dx $$
$$ \int \tilde{f}(x) c_f^{-1} \frac{\tilde{g}^*(x)}{\tilde{f}(x)} \log g_\theta(x) dx \geq \int \tilde{f}(x) c_f^{-1} \frac{\tilde{g}^*(x)}{\tilde{f}(x)} \mathcal{L}(x) dx $$
$$ \int c_f^{-1} \tilde{g}^*(x) \log g_\theta(x) dx \geq \int c_f^{-1} \tilde{g}^*(x) \mathcal{L}(x) dx $$
$$ \int \frac{c_f^{-1}}{c_g^{-1}} c_g^{-1} \tilde{g}^*(x) \log g_\theta(x) dx \geq \int \frac{c_f^{-1}}{c_g^{-1}} c_g^{-1} \tilde{g}^*(x) \mathcal{L}(x) dx $$
$$ \int \frac{c_f^{-1}}{c_g^{-1}} g^*(x) \log g_\theta(x) dx \geq \int \frac{c_f^{-1}}{c_g^{-1}} c_g^{-1} g^*(x) \mathcal{L}(x) dx $$
$$ \frac{c_f^{-1}}{c_g^{-1}} E_{g^*(x)} \left[ \log g_\theta(x) \right] \geq \frac{c_f^{-1}}{c_g^{-1}} E_{g^*(x)} \left[ \mathcal{L}(x) \right] $$
This was just glossed over by the proposed lower bound but it is important to highlight, especially since the authors highlight that beta-VAEs don't optimize a lower bound to the expected log marginal likelihood.

In section 4, the authors introduce a pre-initialization procedure to mitigate posterior collapse, which I think is great to include, but isn't well-motivated. Posterior collapse is a problem, but in the VampPrior paper, the authors showed empirical evidence that using the VampPrior mitigates posterior collapse in hierarchical deep generative models, which are much more prone to posterior collapse compared to standard deep generative models. Thus, did the authors experience posterior collapse and they found this pre-initialization procedure mitigated this? Empirical evidence is needed to demonstrate that this procedure mitigates posterior collapse, or at the very least leads to improvement gains compared to not using the pre-initialization procedure.

In section 5.1.2 the authors state the procedure used in Wang et al., 2019. leads to a biased IS estimator as the estimated pdf values appear in the denominator; they then propose an alternative that "retains the convenient statistical properties of the IS estimator". I don't think this is a fair criticism. While it is true that the approach used by Wang et al., 2019 leads to a biased IS estimator of $\frac{f(x)}{g_\theta(x)}$ the proposed approach is also a biased estimator of this ratio. The proposed approach leads to an unbiased estimate of $\frac{f(x)}{g_\theta^M(x)}$ but as $g_\theta^M(x)$ is an estimator of $g_\theta(x)$, it has the same problem as Wang et al., 2019. While a priori, I expect the proposed approach would lead to lower variance, both are biased.

Also, it isn't clear how this algorithm is combined with AIS. Is the VAE retrained every AIS iteration using samples from the previous iteration? Is the VAE retrained from scratch or is warm-started? The paper would greatly benefit from an algorithm table.

Lastly, my biggest concern is the results section. First, the lack of comparisons is not acceptable. This is especially true given that the proposed approach is very similar to Wang et al., 2019. I think at the minimum, all results should be compared against 1) Wang et al., 2019 2) the proposed method without the initialization procedure, and 3) any AIS algorithm that adapts the proposal by optimizing some form of loss function, i.e. [3]. Second, error bars are missing for the first 2 experiments. Third, in section 5.2.2, the authors state that the KL can't be computed due high values of the pdf; I'm confused by this statement. What is high in values, the target? Is it forward KL or reverse KL? If it is forward KL, then I don't see how the target being high in value (assuming that it is the target) would lead to a numerically unstable computation of KL. Fourth, the combination of proposed VAE with cross-entropy minimization for failure probabilities is non-trivial and I think at the very least needs a section in the appendix. For example, in the duffing oscillator, what is the likelihood function?

# References
[1] Importance Weighted Autoencoders. https://arxiv.org/abs/1509.00519
[2] Variational Sequential Monte Carlo. https://proceedings.mlr.press/v84/naesseth18a/naesseth18a.pdf
[3] Stochastic Gradient Population Monte Carlo. https://ieeexplore.ieee.org/ielaam/97/8966529/8903438-aam.pdf
[4] Policy Gradient Importance Sampling for Bayesian Inference. https://ieeexplore.ieee.org/abstract/document/9468927

---

> ### Author Response · Authors · 2023-11-30
> **Response to Reviewer VqJg part 1**
>
> Dear Reviewer, we are pleased to submit a new version of our paper. We were able to address all your comments. Overall, this leads to a significant improvement of the paper, and we hope that you will be satisfied with this new version. We answer below to your remarks and we comment the modifications made. Of course, we are at your disposal should you require further information.
>
> We added a background section (Section 2) in which we recall the general principle of importance sampling, the main challenges of high-dimensional importance sampling and the use of neural networks with and for importance sampling. We fixed the formatting issue pointed out, thank you.
>
> > Next, the weighted ELBO in equation 12 is not a lower bound of the expected log marginal likelihood, nor does it lead to the same gradient as the lower bound in eq 11
>
> Since both normalising constants $c\_f$ and $c\_g$ do not depend on the parameters to optimise $\boldsymbol{\phi}$, $\boldsymbol{\theta}$ and $\boldsymbol{\lambda}$, it is equivalent to maximise $\mathbb{E}\_{g^*}\left[\log\left(g\_{\boldsymbol{\theta}}\left(\mathbf{X}\right)\right)\right]$ and $\frac{c\_f^{-1}}{c\_g^{-1}}\mathbb{E}\_{g^*}\left[\log\left(g\_{\boldsymbol{\theta}}\left(\mathbf{X}\right)\right)\right]$. Moreover, as remarked by the Reviewer, the new objective function of the VAE with weighted samples $\text{wELBO}\left(\boldsymbol{\phi},\boldsymbol{\theta},\boldsymbol{\lambda}\right)$ defined in Equation (13) is a lower bound of $\frac{c\_f^{-1}}{c\_g^{-1}}\mathbb{E}\_{g^*}\left[\log\left(g\_{\boldsymbol{\theta}}\left(\mathbf{X}\right)\right)\right]$. Therefore, we have: \begin{equation}
>      \dfrac{c\_f}{c\_g}\text{wELBO}\left(\boldsymbol{\phi},\boldsymbol{\theta},\boldsymbol{\lambda}\right) \leq \mathbb{E}\_{g^*}\left[\log\left(g\_{\boldsymbol{\theta}}\left(\mathbf{X}\right)\right)\right].
>  \end{equation} As in the classical VAE framework, we train the VAE by maximising the tractable lower bound $\frac{c\_f}{c\_g}\text{wELBO}\left(\boldsymbol{\phi},\boldsymbol{\theta},\boldsymbol{\lambda}\right)$. But once more, since both constants do not depend on the parameters to optimise, it is equivalent to maximise $\text{wELBO}\left(\boldsymbol{\phi},\boldsymbol{\theta},\boldsymbol{\lambda}\right)$. To sum up, we can consider that the proposed objective function $\text{wELBO}\left(\boldsymbol{\phi},\boldsymbol{\theta},\boldsymbol{\lambda}\right)$ is a lower bound of $\mathbb{E}\_{g^*}\left[\log\left(g\_{\boldsymbol{\theta}}\left(\mathbf{X}\right)\right)\right]$ up to a multiplicative constant, but has the advantage to be independent from the unknown constants $c\_f$ and $c\_g$. At last, one can remark that if $c\_f = c\_g = 1$, then $\text{wELBO}\left(\boldsymbol{\phi},\boldsymbol{\theta},\boldsymbol{\lambda}\right)$ is a real lower bound of $\mathbb{E}\_{g^*}\left[\log\left(g\_{\boldsymbol{\theta}}\left(\mathbf{X}\right)\right)\right]$.
>
> > especially since the authors highlight that beta-VAEs don't optimize a lower bound to the expected log marginal likelihood.
>
> About the Reviewer's remark about $\beta$-VAEs, we recall the expression of the objective function when considering a $\beta$-VAE:\begin{equation}
> \text{ELBO}\_{\beta}\left(\boldsymbol{\phi},\boldsymbol{\theta}\right) = \mathbb{E}\_{g^*}\left[\mathbb{E}\_{q\_{\boldsymbol{\phi}}\left(.|\mathbf{X}\right)}\left[\log\left(g\_{\boldsymbol{\theta}}\left(\mathbf{X}|\mathbf{Z}\right)\right)\right]\right] - \beta\mathbb{E}\_{g^*}\left[D\_{\text{KL}}\left(q\_{\boldsymbol{\phi}}\left(\mathbf{Z}|\mathbf{X}\right)\Vert p\left(\mathbf{Z}\right)\right)\right] \mbox{ with } \beta \in [0,1].
>  \end{equation}If $\beta\leq 1$, since the Kullback-Leibler divergence is non-negative, we have $\text{ELBO}\_{\beta}\left(\boldsymbol{\phi},\boldsymbol{\theta}\right)\geq \text{ELBO}\left(\boldsymbol{\phi},\boldsymbol{\theta}\right)$, and if $\beta<1$, we have no guarantee that $\text{ELBO}\_{\beta}\left(\boldsymbol{\phi},\boldsymbol{\theta}\right)$ is still a lower bound of $\mathbb{E}\_{g^*}\left[\log\left(g\_{\boldsymbol{\theta}}\left(\mathbf{X}\right)\right)\right]$, even up to a multiplicative constant. We add a sentence to make this point clearer after the introduction of $\text{wELBO}\left(\boldsymbol{\phi},\boldsymbol{\theta},\boldsymbol{\lambda}\right)$.

---

> > ### Author Response · Authors · 2023-11-30
> > **Response to Reviewer VqJg part 2**
> >
> > > In section 4, the authors introduce a pre-initialization procedure to mitigate posterior collapse, which I think is great to include, but isn't well-motivated. Posterior collapse is a problem, but in the VampPrior paper, the authors showed empirical evidence that using the VampPrior mitigates posterior collapse in hierarchical deep generative models, which are much more prone to posterior collapse compared to standard deep generative models. Thus, did the authors experience posterior collapse and they found this pre-initialization procedure mitigated this? Empirical evidence is needed to demonstrate that this procedure mitigates posterior collapse, or at the very least leads to improvement gains compared to not using the pre-initialization procedure.
> >
> > When developing the proposed algorithm for adaptive IS with a VAE, we wanted the method to be general enough to be adapted to all kinds of problems, such as multimodal problems for example. To do so, we first tried to apply a classical VAE with a standard Normal prior and without pre-training on a two-modal problem. However, with that setting, the VAE did not caught both modes and we experienced posterior collapse. Moreover, we also applied the procedure using VampPrior but without any pre-training procedure, but we faced the same problem. We also tried to use $\beta$-VAEs but results were not satisfying. Therefore, we decided to try a new approach by combining VampPrior and the proposed pre-training procedure and results were much better. A first illustration of the interest of the use of VampPrior and of the pre-training procedure is provided in Figure 4b. One can clearly see in the latent space that the four modes of the multimodal problem have been well caught by the VAE thanks to the whole procedure. In addition, we will motivate and provide empirical evidence of the interest of the procedure on a new multimodal example and additional experiments. We will show on it that without both VampPrior and the pre-training procedure, results are not as good as with both improvements.
> >
> > > In section 5.1.2 the authors state the procedure used in (Wang et al., 2019) leads to a biased IS estimator as the estimated pdf values appear in the denominator; they then propose an alternative that "retains the convenient statistical properties of the IS estimator". I don't think this is a fair criticism.
> >
> > As motivated earlier, it is crucial for our method to have access to the PDF of the built distribution parameterised by a VAE, and it is not straightforward. A naive way to do so have been proposed in (Wang et al. (2019)) and consists in estimating the PDF value of the resulting distribution $g\_{\boldsymbol{\theta}}$ with the procedure described at the beginning of the new Section 6.1. However, when performing the estimation of an expectation, this method leads to a biased estimator $\widehat{I}\_{g\_{\boldsymbol{\theta}},N}^{\text{IS}}$ since the estimated values of the PDF appear in the denominator. To overcome this issue, we propose no longer to estimate the PDF values of $g\_{\boldsymbol{\theta}}$ but to replace the whole distribution $g\_{\boldsymbol{\theta}}$ by an estimate $g\_{\boldsymbol{\theta}}^M$ for which we can compute exactly and straightforwardly the PDF. Indeed, by construction of the VAE, we have:\begin{equation}
> >     g\_{\boldsymbol{\theta}}\left(.\right) = \int\_{\mathcal{Z}}g\_{\boldsymbol{\theta}}\left(.|\mathbf{z}\right)p\left(\mathbf{z}\right)d\mathbf{z}.
> > \end{equation} Then, based on this expression, we introduce an estimate of the whole distribution $g\_{\boldsymbol{\theta}}$ given by the mixture of distributions: \begin{equation}
> >     g\_{\boldsymbol{\theta}}^M\left(.\right) = \dfrac{1}{M}\sum\_{m=1}^Mg_{\boldsymbol{\theta}}\left(.\left|\mathbf{Z}^{(m)}\right.\right),
> > \end{equation} with $M\geq 1$ and $\left(\mathbf{Z}^{(m)}\right)\_{m\in[[1,M]]}\in\mathcal{Z}^M$ an i.i.d. sample distributed according to the prior $p$. The main point of this new procedure is that we no longer work with $g\_{\boldsymbol{\theta}}$ at all but only with the distribution $g\_{\boldsymbol{\theta}}^M$. Indeed, the IS estimator used in the following sections is: \begin{equation}
> >     \widehat{I}\_{g\_{\boldsymbol{\theta}}^M,N}^{\text{IS}} = \dfrac{1}{N}\sum\_{n=1}^N\psi\left(\mathbf{X}^{(n)}\right)\dfrac{f\left(\mathbf{X}^{(n)}\right)}{g\_{\boldsymbol{\theta}}^M\left(\mathbf{X}^{(n)}\right)},
> > \end{equation} with $\left(\mathbf{X}^{(n)}\right)\_{n\in[[1,N]]}\in\mathcal{X}^N$ an i.i.d. sample distributed according to $g\_{\boldsymbol{\theta}}^M$. At last, we can verify that $\widehat{I}\_{g\_{\boldsymbol{\theta}}^M,N}^{\text{IS}}$ is an unbiased and convergent estimator of $I$. A detailed proof has been added as an appendix in the article.

---

> > > ### Author Response · Authors · 2023-11-30
> > > **Response to Reviewer VqJg part 3**
> > >
> > > > Also, it isn't clear how this algorithm is combined with AIS. Is the VAE retrained every AIS iteration using samples from the previous iteration? Is the VAE retrained from scratch or is warm-started? The paper would greatly benefit from an algorithm table.
> > >
> > > To make clearer how we integrate the VAE in an adaptive IS algorithm, we added the two corresponding pseudo-codes in the article: Algorithms 2 and 3 in the revised paper.
> > >
> > > > Lastly, my biggest concern is the results section. First, the lack of comparisons is not acceptable. This is especially true given that the proposed approach is very similar to Wang et al., 2019. I think at the minimum, all results should be compared against 1) Wang et al., 2019 2) the proposed method without the initialization procedure, and 3) any AIS algorithm that adapts the proposal by optimizing some form of loss function, i.e. [3]. Second, error bars are missing for the first 2 experiments. Third, in section 5.2.2, the authors state that the KL can't be computed due high values of the pdf; I'm confused by this statement. What is high in values, the target? Is it forward KL or reverse KL? If it is forward KL, then I don't see how the target being high in value (assuming that it is the target) would lead to a numerically unstable computation of KL. Fourth, the combination of proposed VAE with cross-entropy minimization for failure probabilities is non-trivial and I think at the very least needs a section in the appendix. For example, in the duffing oscillator, what is the likelihood function?
> > >
> > > 1) We will provide additional simulations to compare our method to the one of (Wang et al., 2019) and also with a classical VAE without any initialisation procedure.
> > > 2) We recall that the expression of the forward Kullback-Leibler divergence between the target distribution $g\_2^*$ and the approximating one $\widehat{g}\_2$ is $D\_{\text{KL}}\left(g\_2^*\Vert g\right) = \mathbb{E}\_{g\_2^*}\left[\log\left(g\_2^*\left(\mathbf{X}\right)\right) - \log\left(\widehat{g}\_2\left(\mathbf{X}\right)\right)\right]$. For estimating this quantity, we first draw an i.i.d. sample $\left(\mathbf{X}^{(n)}\right)\_{n\in[[1,N]]}$ from $g\_2^*$, we compute the log-PDF values for each generated point and we return the empirical mean. However, if for some $n\in[[1,N]]$ the PDF $\widehat{g}\_2\left(\mathbf{X}^{(n)}\right)$ is very very close to $0$, the corresponding log-value $\log\left(\widehat{g}\_2\left(\mathbf{X}^{(n)}\right)\right)$ will be a very large negative value. Then, it might be possible that the software, Python for us, can not handle a such high value and consider it as "Not a number" (NaN). If so, the estimated Kullback-Leibler divergence will also be a NaN.. To overcome this issue, we will investigate the use of other measures to quantify the dissimilarity between the target distribution and the estimated one, such as the kernel distance [1] for example.
> > > 3) We added the pseudo-code of the proposed CE-VAE algorithm: Algorithm 3 in the revised paper.
> > >
> > > Reference:
> > >
> > > [1] Jeff M Philips and Suresh Venkatasubramanian. A gentle introduction to the kernel distance. arXiv preprint arXiv:1103.1625,2011.

---

> > > > ### Comment · Reviewer_VqJg · 2023-12-20
> > > > **Response**
> > > >
> > > > Concerning the integration with the VAE and AIS, my concern has been addressed. The added algorithm table makes things substantially clearer.
> > > >
> > > > Concerning, comparing the results to Wang et al., 2019 and a classical VAE, these results are important to include.
> > > >
> > > > Concerning the KL divergence, I now am on the same page!
> > > >
> > > > The psuedo-code for the CE-VAE is great and substantially helps with readability.

---

> > > ### Comment · Reviewer_VqJg · 2023-12-20
> > > **Response to part 2**
> > >
> > > Concerning the pre-training procedure, this is fascinating! I think including those results would be *very* useful to the target audience!
> > >
> > > Concerning the biased IS estimator issue, my issue has been addressed. I was mistaken; thank you so much for the proof! I highly suggest rewriting eq 1in section 6.1 to make things easier to read.

---

> > ### Comment · Reviewer_VqJg · 2023-12-20
> > **Response part 1**
> >
> > I agree that the minimum obtained by the proposed procedure is the same as minimum obtained by minimizing the lower-bound in equation 11. The added sentence in the paper has addressed my original issue which was that the w-ELBO is not a lower bound to the log-likelihood nor does it produce the same gradients but it does lead to the same solution.

---

### Review · Reviewer_G7sn · 2023-11-18

**Summary Of Contributions:**

The paper demonstrates how to learn the optimal proposal distribution in importance sampling (IS) using variational autoencoder (VAEs) and use it in adaptive IS. This requires a few key ingredients:
- First, the learning of VAEs is done using weighted samples, for which the paper introduces a new learning objective called weighted ELBO.
- Second, the paper uses VampPrior in VAEs such that the construction of density estimators can be effectively dependent on the prior, making the estimates independent as required by IS.
- Additionally, a new way based on supervised + unsupervised pre-training is introduced to avoid posterior collapse, which is needed as the problems considered as multi-modal for which VAEs can suffer from posterior collapse.
- Finally, the paper shows how to use these together to perform adaptive IS.

Extensive simulations in multiple setups are performed to illustrate, validate and compare the proposed method against established baselines. Some of the limitations (e.g. still suffering from high-dimensional problems) are also discussed.

**Audience:**

Yes

**Claims And Evidence:**

Yes

**Requested Changes:**

Requests
- *critical* See my comments about the unbiasedness. This needs to be clarified and addressed accordingly.
- *good-to-have* See my comments about writing in *Weakness* above. Those can strengthen the paper.

**Strengths And Weaknesses:**

**Strengths**

Overall
- The claims of the paper matches the contents very well.
- The contents of the paper are not only useful for people interested in the particular setting it studies (learning density using weighted samples, IS, failure estimation) but could be a good reading for general audience in probabilistic machine learning to learn how various techniques can be used together in practice.

Writing
- Each part of the paper is very well-written with sufficient background knowledge introduced. I believe this would be very helpful for people who are not familiar with one of the related filed.

Method
- The proposed method is neat and based on a set of new or established techniques that are technically sound.

**Questions**

Method
- Unbiasedness: Does (17) actually lead to unbiased IS? It looks to me that it's still biased because of the use of Monte Carlo + g appearing in the denominator. Its improvement from (16) is mostly the independence. Can the authors clarify on this?

**Weaknesses**

Writing
- Abstract: The abstract reads more like a list of "things we do" (or concrete contributions) that should appear in the end of the introduction. For abstract, I would like to see a more high-level summary/storyline of the challenges and intuitions of the proposed, e.g. starting with why and then what. If the authors agree with my view, perhaps they could rewrite that part of the abstract.
- Overall storyline: The current storyline looks a bit weird to me. In particular, the introduction of VampPrior seems to be not well-motivated, until I see it is needed in later sections. Perhaps the authors should consider bring the technique challenges (e.g. VampPrior is motivated by (17) for indepedence) in section 5 upfront and then motivate individual choices.
- Related work: Although they in the form they are proposed can not work with weighted samples, normalizing flows should at least be mentioned (e.g. in the beginning of section 2) as they are considered as the best performing method for general density estimation.
Experiments
- While I agree that methods based on mixture of Gaussians requires prior knowledge of the number of modes, it's also a valid argument to say one can just pick a larger, conservative number. Maybe that would hurts the performance if the number is too large? If that's true (from new experiments), maybe that's something worth including.

**Misc**

Presentation
- Consider making table 1 to the top instead of in the middle of the main texts.
- Figure 4 and 5 are way too large
- Somehow I have difficulty selecting texts on the PDF file.

---

> ### Author Response · Authors · 2023-11-30
> **Response to Reviewer G7sn part 1**
>
> Dear Reviewer, we are pleased to submit a new version of our paper. We were able to address all your comments. Overall, this leads to a significant improvement of the paper, and we hope that you will be satisfied with this new version. We answer below to your remarks and we comment the modifications made. Of course, we are at your disposal should you require further information.
>
> > Unbiasedness: Does (17) actually lead to unbiased IS? It looks to me that it's still biased because of the use of Monte Carlo + g appearing in the denominator. Its improvement from (16) is mostly the independence. Can the authors clarify on this?
>
> As motivated earlier, it is crucial for our method to have access to the PDF of the built distribution parameterised by a VAE, and it is not straightforward. A naive way to do so have been proposed in (Wang et al. (2019)) and consists in estimating the PDF value of the resulting distribution $g\_{\boldsymbol{\theta}}$ with the procedure described at the beginning of the new Section 6.1. However, when performing the estimation of an expectation, this method leads to a biased estimator $\widehat{I}\_{g\_{\boldsymbol{\theta}},N}^{\text{IS}}$ since the estimated values of the PDF appear in the denominator. To overcome this issue, we propose no longer to estimate the PDF values of $g\_{\boldsymbol{\theta}}$ but to replace the whole distribution $g\_{\boldsymbol{\theta}}$ by an estimate $g\_{\boldsymbol{\theta}}^M$ for which we can compute exactly and straightforwardly the PDF. Indeed, by construction of the VAE, we have:\begin{equation}
>     g\_{\boldsymbol{\theta}}\left(.\right) = \int\_{\mathcal{Z}}g\_{\boldsymbol{\theta}}\left(.|\mathbf{z}\right)p\left(\mathbf{z}\right)d\mathbf{z}.
> \end{equation} Then, based on this expression, we introduce an estimate of the whole distribution $g\_{\boldsymbol{\theta}}$ given by the mixture of distributions: \begin{equation}
>     g\_{\boldsymbol{\theta}}^M\left(.\right) = \dfrac{1}{M}\sum\_{m=1}^Mg\_{\boldsymbol{\theta}}\left(.\left|\mathbf{Z}^{(m)}\right.\right),
> \end{equation} with $M\geq 1$ and $\left(\mathbf{Z}^{(m)}\right)\_{m\in[[1,M]]}\in\mathcal{Z}^M$ an i.i.d. sample distributed according to the prior $p$. The main point of this new procedure is that we no longer work with $g\_{\boldsymbol{\theta}}$ at all but only with the distribution $g\_{\boldsymbol{\theta}}^M$. Indeed, the IS estimator used in the following sections is: \begin{equation}
>     \widehat{I}\_{g\_{\boldsymbol{\theta}}^M,N}^{\text{IS}} = \dfrac{1}{N}\sum\_{n=1}^N\psi\left(\mathbf{X}^{(n)}\right)\dfrac{f\left(\mathbf{X}^{(n)}\right)}{g\_{\boldsymbol{\theta}}^M\left(\mathbf{X}^{(n)}\right)},
> \end{equation} with $\left(\mathbf{X}^{(n)}\right)\_{n\in[[1,N]]}\in\mathcal{X}^N$ an i.i.d. sample distributed according to $g\_{\boldsymbol{\theta}}^M$. At last, we can verify that $\widehat{I}\_{g\_{\boldsymbol{\theta}}^M,N}^{\text{IS}}$ is an unbiased and convergent estimator of $I$. A detailed proof has been added as an appendix in the article.
>
> > Abstract: The abstract reads more like a list of "things we do" (or concrete contributions) that should appear in the end of the introduction. For abstract, I would like to see a more high-level summary/storyline of the challenges and intuitions of the proposed, e.g. starting with why and then what. If the authors agree with my view, perhaps they could rewrite that part of the abstract.
>
> We agree with this remark and we made the corresponding modification in the abstract and in the introduction.

---

> > ### Author Response · Authors · 2023-11-30
> > **Response to Reviewer G7sn part 2**
> >
> > > Overall storyline: The current storyline looks a bit weird to me. In particular, the introduction of VampPrior seems to be not well-motivated, until I see it is needed in later sections. Perhaps the authors should consider bring the technique challenges (e.g. VampPrior is motivated by (17) for indepedence) in section 5 upfront and then motivate individual choices.
> >
> > First of all, we would like to clarify that VampPrior is not related at all with Equation (17). Indeed, whatever the chosen prior $p$, the resulting distribution of the VAE can still be written in the form: \begin{equation}
> >     g\_{\boldsymbol{\theta}}\left(.\right) = \int\_{\mathcal{Z}}g\_{\boldsymbol{\theta}}\left(.|\mathbf{z}\right)p\left(\mathbf{z}\right)d\mathbf{z}.
> > \end{equation} Then, the proposed approximating distribution $g\_{\boldsymbol{\theta}}^M$ in Equation (17), chosen for the reasons described above, is relevant for every choice of the prior $p$ and is not related with VampPrior.\\Moreover, when developing the proposed algorithm for adaptive IS with a VAE, we would like the method to be general enough to be adapted to all kinds of problems, such as multimodal problems for example. We introduce and motivate this idea from a mathematical point of view in Section 3.3 and we also show how to include the new flexible prior in the VAE. Since Section 3 is a general mathematical presentation of the VAEs, we will provide empirical evidence of the interest of the procedure on a new multimodal example and additional experiments. We will show on it that without both VampPrior and the pre-training procedure, results are not as good as with both improvements. A first illustration of the interest of the use of VampPrior and of the pre-training procedure is provided in Figure 4b. One can clearly see in the latent space that the four modes of the four branch problem have been well caught by the VAE thanks to the whole procedure.
> >
> > > Related work: Although they in the form they are proposed can not work with weighted samples, normalizing flows should at least be mentioned (e.g. in the beginning of section 2) as they are considered as the best performing method for general density estimation. Experiments
> >
> > We added at the beginning of the revised article a background section (Section 2) reviewing existing works on high-dimensional importance sampling and importance sampling with neural networks. We mention there normalising flows. However, we do not plan to perform numerical experiments involving normalising flows, as we would like to consider this as outside of the scope of the article.
> >
> > > While I agree that methods based on mixture of Gaussians requires prior knowledge of the number of modes, it's also a valid argument to say one can just pick a larger, conservative number. Maybe that would hurts the performance if the number is too large? If that's true (from new experiments), maybe that's something worth including.
> >
> > We agree that the described procedure could be possible. However, artificially increasing the number of components in the mixture will increase the number of parameters to estimate at each iteration. If this number becomes too large, it will badly affects the performances of the corresponding algorithm. We will provide new experiments in Section 6.3 to illustrate this phenomenon.

---

> > > ### Comment · Reviewer_G7sn · 2023-12-20
> > >
> > > Thanks for the response.
> > > The new estimate looks good to me as being unbiased and consistent.
> > > The revised story also reads better now as the introduction of VampPrior is better motivated as well as the necessary background of IS.

---

> > > > ### Author Response · Authors · 2023-12-20
> > > >
> > > > Thank you for this comment.

---

### Decision · Action_Editor_FuqH · 2023-12-21

**Recommendation:** Accept with minor revision

**Comment:**

All three reviewers found that the authors' rebuttal was convincing and it addressed all of their issues. Thus, they unanimously lean towards acceptance of the paper. Overall, the paper is a nice contribution to TMLR and I second the reviewers' sentiment towards acceptance.

Reviewrer #VqJg mentioned some actionable suggestions (mostly for the experimental section) which would be great to include in the final version of the paper.

**Audience:**

The paper shows how to build a proposal for IS and AIS, using (an extension of) the VAE for that. The reviewers found that the proposed approach is of interest and can lead to cool applications. Thus, the paper fits within TMLR's scope.

**Claims And Evidence:**

The reviewers agree that the claims in the paper are well supported by evidence and experiments.

---

> ### Author Response · Authors · 2024-01-19
> **Submission of the revised article**
>
> This is just a comment to point out we updated the submission with a new revision following reviewers' remarks.